# INCLUDE: EVALUATING MULTILINGUAL LANGUAGE UNDERSTANDING WITH REGIONAL KNOWLEDGE

**Angelika Romanou**[1,*]**, Negar Foroutan**[1]**, Anna Sotnikova**[1]**, Zeming Chen**[1]**,
Sree Harsha Nelaturu**[3,5]**, Shivalika Singh**[2]**, Rishabh Maheshwary**[7]**, Micol Altomare**[6]**,
Mohamed A. Haggag**[8]**, Imanol Schlag**[4] **Marzieh Fadaee**[2,†]**, Sara Hooker**[2,†]**, Antoine Bosselut**[1,†]

[1]EPFL, [2]Cohere For AI, [3]Cohere For AI Community, [4]ETH Zurich, [5]Saarland University,
[6]University of Toronto, [7]ServiceNow, [8]Georgia Institute of Technology

Dataset Contributors:
**Snegha A, Alfonso Amayuelas, Azril Hafizi Amirudin, Viraat Aryabumi, Danylo
Boiko, Michael Chang, Jenny Chim, Gal Cohen, Aditya Kumar Dalmia, Abraham
Diress, Sharad Duwal, Daniil Dzenhaliou, Daniel Fernando Erazo Florez, Fabian
Farestam, Joseph Marvin Imperial, Shayekh Bin Islam, Perttu Isotalo, Maral
Jabbarishiviari, Börje F. Karlsson, Eldar Khalilov, Christopher Klamm, Fajri Koto,
Dominik Krzemiński, Gabriel Adriano de Melo, Syrielle Montariol, Yiyang Nan,
Joel Niklaus, Jekaterina Novikova, Johan Samir Obando Ceron, Debjit Paul, Esther
Ploeger, Jebish Purbey, Swati Rajwal, Selvan Sunitha Ravi, Sara Rydell, Roshan
Santhosh, Drishti Sharma, Marjana Prifti Skenduli, Arshia Soltani Moakhar, Bardia
Soltani Moakhar, Ran Tamir, Ayush Kumar Tarun, Azmine Toushik Wasi, Thenuka
Ovin Weerasinghe, Serhan Yilmaz, Mike Zhang**

## ABSTRACT

The performance differential of large language models (LLM) between languages hinders their effective deployment in many regions, inhibiting the potential economic and societal value of generative AI tools in many communities. However, the development of functional LLMs in many languages (*i.e.*, multilingual LLMs) is bottlenecked by the lack of high-quality evaluation resources in languages other than English. Moreover, current practices in multilingual benchmark construction often translate English resources, ignoring the regional and cultural knowledge of the environments in which multilingual systems would be used. In this work, we construct an evaluation suite of 197,243 QA pairs from local exam sources to measure the capabilities of multilingual LLMs in a variety of regional contexts. Our novel resource, INCLUDE,[1] is a comprehensive knowledge- and reasoning-centric benchmark across 44 written languages that evaluates multilingual LLMs for performance in the actual language environments where they would be deployed.

## 1 INTRODUCTION

The rapid advancement of AI technologies underscores the importance of developing LLMs that are proficient across diverse linguistic and cultural contexts, ensuring fair and equitable performance for stakeholders from various language groups. However, the lack of high-quality evaluation benchmarks in many languages discourages practitioners from training multilingual LLMs to meet this challenge. This evaluation gap limits the effective deployment of LLMs for many regions, exacerbates digital divides, and inhibits the economic and societal value of AI tools in many underserved communities.

The source of this gap is the multitude of challenges in evaluating LLMs for multilingual contexts. First, at a meta-level, the majority of benchmarks for LLMs are only in English (Hendrycks et al., 2020, *inter alia*). While non-English benchmarks exist for some tasks (Singh et al., 2024; Aakanksha et al., 2024; Pozzobon et al., 2024), they usually focus on single languages (Li et al., 2023; Koto et al.,

---

*Corresponding email: `angelika.romanou@epfl.ch`; † Equal Supervision
[1]`https://huggingface.co/datasets/CohereForAI/include-base-44`

2024), specific regions (Cañete et al., 2020; Guevara-Rukoz et al., 2020; Cahyawijaya et al., 2022; Adelani et al., 2024; Etxaniz et al., 2024b), or a particular domain (Wang et al., 2024a), ignoring the importance of joint evaluation to trace and unlock the benefits that multilingual capabilities could bring to low-resource languages (Pfeiffer et al., 2022; Üstün et al., 2024; Aryabumi et al., 2024).

Technical challenges also abound due to the manner in which multilingual datasets are often collected. Certain datasets are constructed using manually applied templates, resulting in low prompt and completion diversity (Muennighoff et al., 2022). Many more are composed of translations from high-resource languages (*e.g.*, English; Lai et al., 2023; Foroutan et al., 2023; Holtermann et al., 2024; Myung et al., 2024). These datasets often contain errors (Ponti et al., 2020; Plaza et al., 2024) and create *translationese artifacts* (Vanmassenhove et al., 2021; Hartung et al., 2023; Savoldi et al., 2021; Ji et al., 2023). Most importantly, they do not accurately reflect the regional and cultural contexts captured by different languages (Aakanksha et al., 2024; Awad et al., 2020; Ramezani & Xu, 2023; Singh et al., 2024). As seen in Figure 1 (a) (Regional Knowledge), a legal question posed in English, Russian, or Greek would likely reflect a user located in a different environment, where different laws may apply to respond correctly. Similarly, also seen in Figure 1 (a) (Cultural Knowledge), historical or cultural perspectives on the same topic may differ among the populaces of different regions.

To resolve this gap, we design a pipeline to collect a large multilingual language understanding benchmark (*i.e.*, INCLUDE) by collecting regional resources (*e.g.*, educational, professional, and practical tests) that are specific to countries and originally created by native speakers of each country's official languages. This collection avoids *translationese* (Bizzoni et al., 2020) and also captures cultural nuances associated with each language, enabling rigorous evaluation of how state-of-the-art models serve diverse language users around the world.

In our experiments, we sample INCLUDE into two subsets for different evaluation budgets and assess an array of closed and open models on these partitions. Our results demonstrate that current models achieve high variance in performance between different languages in INCLUDE, and that models often struggle with questions requiring regional knowledge. Further analysis reveals that models score particularly low on languages on which they are not intentionally trained (*i.e.*, limiting regional knowledge acquisition), and that the possibility of transferring global (*i.e.*, English-aligned) perspectives improves performance for less regional topics across languages.

## 2 PRELIMINARIES: LANGUAGE & KNOWLEDGE

**Language availability.** Languages are typically characterized as a high, medium, or low resource based on reported language availability (Joshi et al., 2020), *i.e.*, the amount of data in a language that is available online. Interestingly, the language availability of documents used for training models (Penedo et al., 2024; Xue, 2020; Conneau et al., 2020; Computer, 2023; Üstün et al., 2024; Singh et al., 2024) differs drastically from the language distribution of non-English LLM benchmarks, with the latter being more scarce. Inspired by this discrepancy, we include 44 languages in our INCLUDE benchmark. In Figure 1, we characterize the included languages based on their reported availability in the mC4 corpus (Xue, 2020), and in Table 8 show further detailed metadata for each language.

**Language represents regional knowledge.** For LLM-based systems to practically useful, they must enable interaction in the preferred languages of their users and be knowledgeable of the environments of those users. We define *regional knowledge* as the specific information, culture, and practices related to a local environment that is relevant for a user's context. However, LLMs such as GPT-4 tend to exhibit a Western bias (Tao et al., 2024) due to the overrepresentation of Western text in training data (AlKhamissi et al., 2024). In INCLUDE, we specifically include questions encompassing the regional and cultural knowledge of a diverse set of high, medium, and low-resource languages.

## 3 THE INCLUDE BENCHMARK

INCLUDE is a dataset of 197,243 MCQA pairs from 1,926 examinations across 44 languages and 15 scripts. These examinations are collected from local sources in 52 countries, representing a rich array of cultural and regional knowledge. All questions in the dataset are presented in their native languages and scripts. In this section, we describe the data collection procedure for INCLUDE, as well as additional categorical labels we assign to each question in the dataset for later analysis.

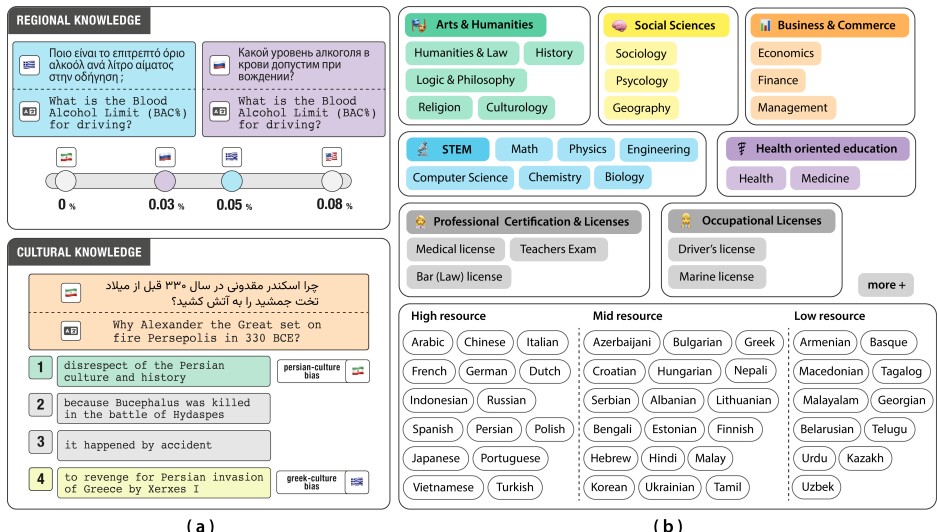

Figure 1: **Overview of INCLUDE**. (a) **Motivation:** Multilingual benchmarks must reflect the cultural and regional knowledge of the language environments in which multilingual LLM systems would be used. (b) **INCLUDE** is a multilingual benchmark compiled from academic, professional, and occupational license examinations reflecting regional and cultural knowledge in 44 languages.

## 3.1 DATA COLLECTION

To construct **INCLUDE**, we collect sources of multiple-choice exams in collaboration with native speakers and regional associations. We primarily focused on three types of exams:

**Academic Exams:** Exams from a variety of subjects (*e.g.*, Humanities, STEM, etc.) at different levels (*e.g.*, middle & high school, university), including country-specific national entrance exams.

**Professional Certifications & Licenses:** Exams issued by industry-specific regulatory bodies for specialized fields, *e.g.*, licensing exams for areas such as legal and medical practice.

**Regional Licenses:** Exams administered by regional authorities that assess specific qualifications, such as driving and marine licenses.

We design **INCLUDE** to assess multilingual capabilities that span beyond academic knowledge to cultural and region-specific understanding. Our data collection focuses on license and certification exams that capture regional knowledge of specific countries (in their official languages), and non-translated academic content from the humanities and social sciences to capture cultural knowledge.

From the collected sources, we extract the multiple-choice questions with their corresponding options and correct answers. More specifically, as this data came in different formats (*e.g.*, PDFs, Javascript HTML forms), we use multiple pipelines to extract QA samples from these sources and curate them in a machine-readable manner. The goal of this stage was to automate data extraction and then rely on human evaluation for verification and feature annotation.

**Quality Control with Native Speakers.** After automatic extraction, we provide native speakers (co-authors in this work) with parsed multiple-choice questions to ensure they were extracted correctly from source documents. In cases of extraction mistakes, annotators manually corrected parsed questions and answer options. Annotators also filtered out samples that referred to images or tables, and verified that samples that rely on additional context (*e.g.*, reading comprehension) include the reference text in the question field. Finally, annotators labeled each question with additional exam metadata, such as the language of the MCQ, its topic in both English and the original language, the academic level (if relevant), and the country of origin. In total, we parsed and verified 118,606 samples across 1,926 exam sources, amounting to 60.2% of the total data in **INCLUDE**.

**Rounding out the Benchmark.** To complete our benchmark, we also consolidate existing datasets with extensive domain coverage in single non-English languages: ArabicMMLU (Koto et al., 2024), ChineseMMLU (Li et al., 2023), TurkishMMLU (Yüksel et al., 2024), PersianMMLU (Ghahroodi

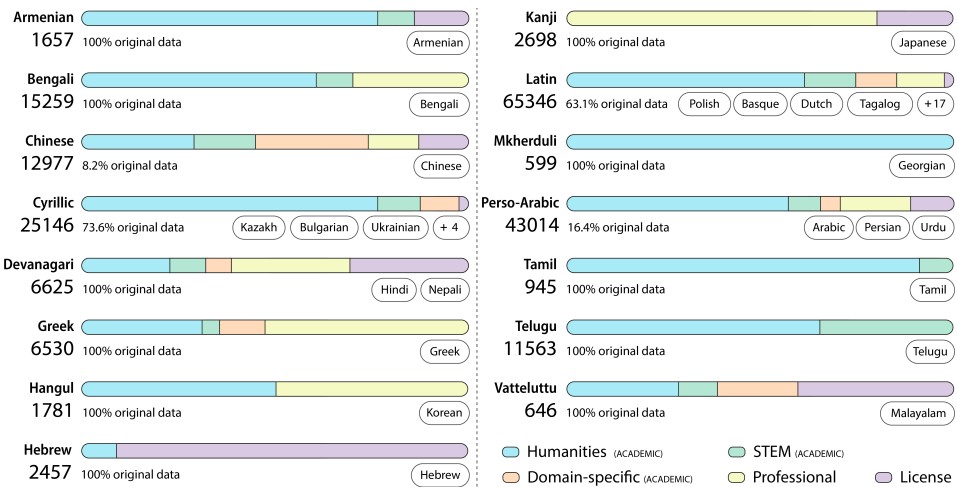

Figure 2: **Overview of the collected data grouped by script.** We depict the languages associated with each script, the total samples in each script, and the percentage of the samples that were collected from new sources that have not been published by the community yet.

et al., 2024) and VNHSGE (Dao et al., 2023), as well as a multilingual benchmark with limited domain coverage across multiple European languages: EXAMS (Hardalov et al., 2020). We repurpose 78,637 samples from these published benchmarks, amounting to 39.8% of the data in **INCLUDE**.

In sum, **INCLUDE** is the largest collection of multilingual exam data to-date. It is composed of 197,243 QA pairs from both novel and existing sources, and covers examinations from more than 1,926 exams in 44 different languages, 15 scripts, and 58 knowledge domains. Figure 1 and Appendix Table 8 summarize the language, domain, and knowledge diversity of the **INCLUDE** benchmark.

## 3.2 CATEGORIZING KNOWLEDGE

The language breadth of **INCLUDE** provides the opportunity to investigate what factors drive multilingual performance. Consequently, we annotate **INCLUDE** samples with category labels corresponding to factors such as the topic of a question and its region-specificity. Given the prohibitive cost of performing sample-level annotation, we only perform a coarse annotation by labeling the exam sources of QA pairs, rather than individual samples. We describe our categorization schemes below.

**Academic Domain.** We manually categorized 1,926 unique exams, following the methodology in Hendrycks et al. (2020). Our categorization follows a two-level taxonomy: a high-level academic area (*e.g.*, Humanities), and particular academic field within this area (*e.g.*, History, Philosophy, Literature).[2] Each exam is categorized based on its title, which indicates its topic (*e.g.*, Greek History) and associated level (*e.g.*, high school, undergraduate, professional certification). Figure 7 provides a breakdown of the number of exam samples per language, organized by this taxonomy.

**Regionality.** To account for regional knowledge, we categorize exam questions into two major groups: **region-agnostic** and **region-specific** knowledge. Region-agnostic questions do not require knowledge of particular regions (*e.g.*, mathematics, physics), and their answers should remain common regardless of the language in which a question is posed. In total, 34.4% of all questions collected were classified as region-agnostic. In contrast, region-specific questions require knowledge that may depend on a particular cultural or geographical context. This category is further divided into three sub-categories:

*Explicitly Regional:* A question is classified as *region-explicit* when it pertains to legal, regulatory, or procedural knowledge of regions. Examples include questions about local laws, certifications, clinical guidelines, or licensing requirements (see Figure 1(a)). 18.8% of all questions were *region-explicit*.

*Cultural:* Language often serves as an implicit marker of culture. For instance, in Figure 1(a), the answer to a question about historical figures in a Greek exam may reflect a different perspective than a similar question posed for a Persian exam. We categorize questions as *cultural* when they pertain to

---

[2]This taxonomy is adapted from the *Outline of Academic Disciplines* found on Wikipedia.

a region's cultural or historical context. This category includes questions for subjects inherently tied to a region's language, history, or social norms. 16.4% of questions were classified as *cultural*.

*Implicitly Regional:* Finally, the *region-implicit* category is a catch-all for other questions whose answers may depend on a certain degree of regional knowledge understanding. These questions are not explicitly regional or culture-related, but may require regional context to answer correctly. For example, business practices may be different depending on region, even if the underlying theory is common in many places. In total, 30.4% of all questions collected were classified as region-implicit.

Detailed annotation procedures for these categories are described in Appendix A.4, and general statistics about regional labels per academic area and academic field are provided in Figure 7.

## 4 EXPERIMENTAL SETUP

In this section, we describe our experimental settings for evaluating models on **INCLUDE**.

### 4.1 DATA SELECTION

The breadth of **INCLUDE** (197,243 QA pairs in 44 languages) makes it amenable to many evaluation use cases, including monolingual evaluation in 44 languages. However, for multilingual evaluation, this same scale is prohibitively expensive for many researchers.[3] Consequently, we curate two subsets of **INCLUDE** for benchmarking multilingual LLMs in different resource settings.

**INCLUDE-BASE**: This subset uniformly samples 22,635 QA pairs (~12% of **INCLUDE**) across languages, knowledge tasks, and academic levels. The goal of this subset is to develop a multilingual benchmark with broad language and task coverage. Each language has a maximum of 550 samples, with 500 drawn from domains that correspond to regional knowledge and 50 from STEM subjects.

**INCLUDE-LITE**: A lightweight subset, uniformly drawn from **INCLUDE-BASE**, designed for rapid assessment of multilingual LLMs with a total of 10,770 samples (~6% of **INCLUDE**). The upper limit per language is 250 samples and only includes region-specific domains.

For standardization (and alignment with prior benchmarks; Hendrycks et al., 2020), **INCLUDE-BASE** and **INCLUDE-LITE** contain only multiple-choice questions with four answer options. Questions from **INCLUDE** with fewer than four options were omitted during sampling, and questions with more than four options were pruned of options until only four remained. In the following sections, we benchmark models (§5.1) and perform analysis (§5.2-5.3) on **INCLUDE-BASE** and **INCLUDE-LITE**.

### 4.2 MODELS

We assess **INCLUDE** on GPT-4o (Achiam et al., 2023) as a state-of-the-art multilingual and general-purpose model. We also investigate the role of scaling by benchmarking models that self-report parameters, comparing the larger Llama-3.1-70B-Instruct (Dubey et al., 2024), the Aya-expanse-32B (Aryabumi et al., 2024), and the Qwen2.5-14B (Yang et al., 2024) 14-billion parameter model with Llama-3.1-Instruct-8B (Dubey et al., 2024), Aya-expanse-8B (Aryabumi et al., 2024) and Qwen2.5-7B. Additionally, we benchmark Mistral-7B (Jiang et al., 2023) and Gemma-7B (Team et al., 2024) along with their Instruct variants. We note that some of the evaluated models neither explicitly claim to support multiple languages nor disclose the languages they were pretrained on. However, in practice, they are heavily adopted in multilingual use cases relative to explicitly multilingual models. Furthermore, even reportedly multilingual models (*e.g.*, Aya-Expanse, which supports 23 languages) do not support all 44 languages included in our benchmark.

**Prompting.** Following Hendrycks et al. (2020), we report both 5-shot and zero-shot scores. For the zero-shot setting, we employ a Chain-of-Thought (CoT; Wei et al., 2022) approach by appending the translation of "let's think step by step" to the prompt (Kojima et al., 2022). We evaluate models using both (1) *In-Language (IL) Prompts*, which present the prompt instructions in the same language as the sample, and (2) *English (Eng.) Prompts*, which provide the prompt instructions in English. For both prompting language settings, we also test a setting where we add a *Regional (Reg.)* prefix

---

[3]The cost of evaluating **INCLUDE** using GPT-4o with 5-shot demonstrations exceeded $1000.

| Model | # Langs | INCLUDE-LITE | | | | INCLUDE-BASE | | | |
|---|---|---|---|---|---|---|---|---|---|
| | | IL Prompt | Eng. Prompt | Reg. + IL Prompt | Reg. + Eng. Prompt | IL Prompt | Eng. Prompt | Reg. + IL Prompt | Reg. + Eng. Prompt |
| **GPT-4o** | - | | | | | | | | |
| - 5-shot | | 77.1 | 76.2 | 76.3 | 76.3 | 77.3 | 76.3 | 76.2 | 76.2 |
| - Zero-shot CoT | | **78.2** | **78.4** | **77.7** | **77.8** | **79.0** | **78.9** | **77.6** | **78.5** |
| **Llama-3.1-70B-Inst.** | - | | | | | | | | |
| - 5-shot | | 70.5 | 70.4 | 70.6 | 70.6 | 70.6 | 70.7 | 70.6 | 70.6 |
| - Zero-shot CoT | | 60.6 | 55.3 | 60.2 | 55.4 | 60.6 | 56.0 | 60.6 | 55.6 |
| **Aya-expanse-32B** | 23 | | | | | | | | |
| - 5-shot | | 52.6 | 57.2 | 49.0 | 60.0 | 52.4 | 56.6 | 49.7 | 60.0 |
| - Zero-shot CoT | | 50.6 | 57.1 | 52.5 | 58.0 | 51.4 | 57.7 | 52.9 | 57.8 |
| **Qwen2.5-14B** | 22 | | | | | | | | |
| - 5-shot | | 60.9 | 61.3 | 60.9 | 60.8 | 61.4 | 61.7 | 61.1 | 61.0 |
| - Zero-shot CoT | | 46.8 | 50.7 | 46.5 | 51.4 | 47.3 | 51.0 | 47.1 | 51.6 |
| **Aya-expanse-8B** | 23 | 37.6 | 46.3 | 38.1 | 48.0 | 37.2 | 46.0 | 37.9 | 47.8 |
| **Mistral-7B (v0.3)** | - | 44.0 | 45.0 | 44.0 | 45.2 | 43.3 | 44.9 | 43.8 | 45.0 |
| **Mistral-7B-Inst. (v0.3)** | - | 43.5 | 44.6 | 44.2 | 44.7 | 43.6 | 44.5 | 44.2 | 44.7 |
| **Gemma-7B** | - | 54.4 | 54.9 | 54.3 | 54.9 | 54.5 | 54.9 | 54.2 | 54.7 |
| **Gemma-7B-Inst.** | - | 39.2 | 40.2 | 38.7 | 39.7 | 38.7 | 39.7 | 38.1 | 39.2 |
| **Qwen2.5-7B** | 22 | 53.4 | 54.8 | 53.3 | 54.2 | 54.1 | 55.2 | 54.0 | 54.5 |
| **Qwen2.5-7B-Inst.** | 22 | 53.4 | 54.2 | 52.8 | 53.7 | 53.8 | 54.6 | 53.2 | 53.9 |
| **Llama-3.1-8B** | - | 50.9 | 52.3 | 50.9 | 51.9 | 51.0 | 51.8 | 51.0 | 51.6 |
| **Llama-3.1-8B-Inst.** | - | 53.4 | 54.8 | 52.7 | 53.4 | 53.4 | 54.6 | 53.0 | 54.4 |

Table 1: Results on **INCLUDE-LITE** and **INCLUDE-BASE**. **In-language Prompt (IL)** reports model accuracy when the prompt instructions are presented in the same language as the sample. **English Prompt (Eng.)** reports model accuracy when the prompt instructions are provided in English. **In-language Regional Prompt (Reg. + IL)** reports model accuracy when a regional prefix is added to the In-language Prompt. **English Regional Prompt (Reg. + Eng.)** reports model accuracy when a regional prefix is added to the English Prompt. **# Langs** reports the number of languages from **INCLUDE** publicly reported to be *intentionally* included in the pretraining data of each model.

to the (either in-language or English) prompt to explicitly contextualize the region and language of the sample, asking the models to consider the cultural and regional characteristics. The maximum generation lengths for the 5-shot and zero-shot CoT configurations are set to 512 and 1024 tokens. Finally, we additionally evaluate **INCLUDE** using the Harness-Eval framework (Gao et al., 2024) using both *In-Language Prompts* and *English Prompts*.

## 5 RESULTS & ANALYSIS

### 5.1 GENERAL PERFORMANCE

Table 1 shows the performance of all models evaluated across the 44 languages in **INCLUDE-BASE**. For larger models, *e.g.*, GPT-4o, Llama-3.1-70B-Instruct, Aya-expanse-32B and Qwen2.5-14B, we provide results for both 5-shot and zero-shot CoT, and report only 5-shot accuracy for other models.

Among all models, GPT-4o achieves the highest performance, reaching an accuracy of ∼77.1% across all domains and examples. We observe that CoT prompting moderately enhances GPT-4o's performance, particularly in Professional and STEM-related exams (Table 2), where the most substantial improvements were seen. In contrast, the smallest gains were observed in exams related to Licenses and the Humanities. Drawing on prior studies that compare CoT and non-CoT prompting strategies across different domains (Sprague et al., 2024), we hypothesize that this observation is due to reasoning skills required in professional examinations (*e.g.*, medicine, law), and computation-heavy subjects in STEM. In contrast, we observe significant performance drops by Llama-3.1-70B-Instruct and Qwen2.5-14B when using CoT prompting on **INCLUDE-BASE**, with the largest drops on subjects involving more mathematical reasoning (Table 2).

For smaller models (≤8B parameters), Gemma-7B delivers the best overall performance, with Qwen2.5-7B and Qwen2.5-7B-Instruct closely behind. While Gemma-7B excels in the Humanities and Licenses categories, the Qwen models surpass others in STEM, Applied Sciences, and Professional domains (Table 2). When comparing models from the same family across different scales, we observe that the Aya-expanse-32B model outperforms the 8B model by ∼12% on average, while the Qwen2.5-14B model shows an average ∼7% improvement over its 7B counterpart on

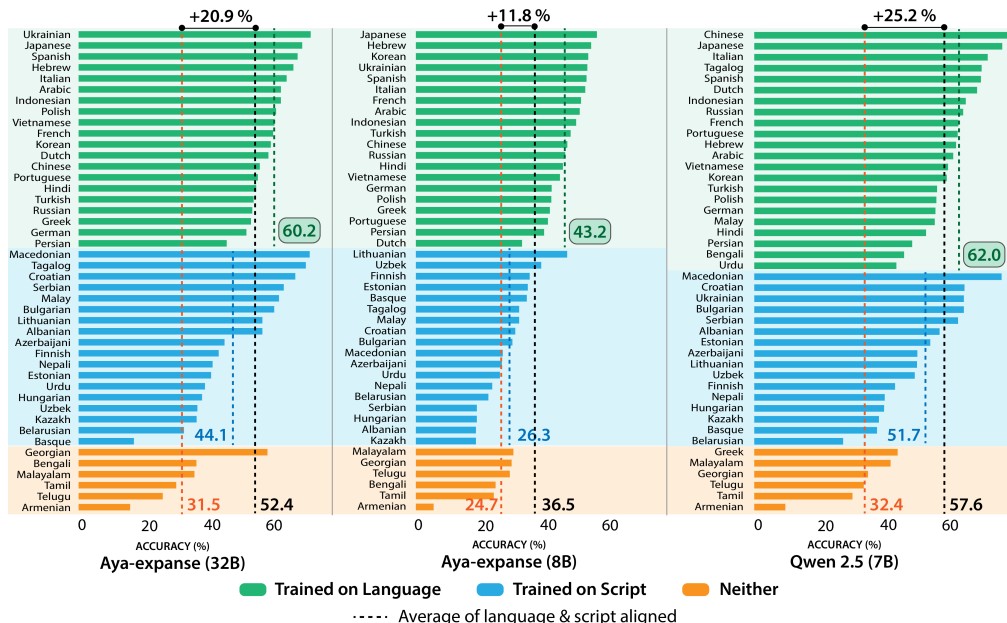

Figure 3: **Performance of models stratified by language using in-language prompting.** Results are grouped by whether the language was explicitly included in the pretraining dataset of the model (**Trained on Language**), whether a similar language with the same script was in the pretraining corpus (**Trained on Script**), or whether there was no linguistically similar language in the pretraining corpus (**Neither**). Color dotted lines represent average performance for each category for a particular model. Black dotted lines represent average performance across all script-aligned languages.

**INCLUDE-BASE** across prompting settings. As the pretraining data remained consistent across the different sizes within both the Aya-expanse and Qwen2.5 model families, we conclude that, with similar training data, increasing model size significantly enhances multilingual capabilities.

Interestingly, we see little benefit to instruction-tuning for improving performance on **INCLUDE-BASE**. Most instruction-tuned models perform slightly worse or on par with their base counterpart, with an outlier performance drop of ∼15% across prompting settings for Gemma-7B-Instruct. A possible explanation for this gap is that instruction-tuned models may have been post-trained predominantly on English data, potentially diminishing multilingual capabilities acquired during pretraining.

Overall, we observe minimal performance differences across the various prompting settings, except in the CoT setting where Qwen2.5-14B and Aya-expanse benefit significantly from English prompts (over the in-language variants) and Llama-3.1-70B-Instruct performs better with in-language prompts. Outside of these exceptions, English prompts offer modest performance improvements for most models, but this improvement remains within 1–2% of in-language prompt performance. Apart from Aya-expanse-8B, specifying the region of the question in the prompt prefix does not generally improve performance, likely because the language of the question serves as an implicit proxy of the relevant region of the question for most languages in our benchmark.

Finally, we observe that performance on **INCLUDE-LITE** is nearly equivalent to **INCLUDE-BASE** across all models, with differences within 1%, demonstrating its suitability for resource-constrained evaluation settings. Likewise, when using the Harness evaluation framework, model performance remains consistent, with results also within 1% (Table 4).

## 5.2 LANGUAGE ANALYSIS

To better understand how LLMs perform on questions in languages seen and unseen during pretraining, we take a deeper look into three open models, *i.e.*, Aya-expanse-8B, Aya-expanse-32B, and Qwen2.5-7B, for which we have details surrounding pretraining data (and its associated language distribution). In this analysis, we specifically test three language exposure scenarios: performance on languages the

model has been *intentionally*[4] trained on, performance on languages the model was not reported to be trained on but for which the corresponding script was reported to be trained on, and performance on completely unseen languages and scripts during pretraining.

Figure 3 presents the language-stratified performance of these models on **INCLUDE-BASE**. As expected, the models demonstrate better performance on languages that were reported as part of their pretraining data (**Trained on Language**). All models also demonstrate some degree of knowledge transfer to languages they were not trained on but which share the same script as languages in their pretraining data (**Trained on Script**). In this scenario, Aya-expanse-32B achieves $44.1\%$ accuracy, while Qwen2.5-7B reaches $51.7\%$ accuracy, aligning with previous research that suggests shared scripts enable cross-lingual transfer between languages (Muller et al., 2021; Xhelili et al., 2024). Other factors may also contribute to each model's performance on unseen languages, though, such as cross-lingual transfer across topologically-similar languages. For example, the presence of Turkish data may enhance the model's performance on Azerbaijani (Senel et al., 2024). Pretraining data contamination, where languages that were not intended to be in the pretraining data may still be incidentally included (Blevins & Zettlemoyer, 2022), may also contribute to these transfer results.

Lastly, we observe that all three models perform poorly on languages whose scripts were not represented in the pretraining corpus (**Neither** in Figure 3) with the exception of the Georgian language for the Aya-expanse-32B model. Data contamination is also less of a confounding factor in these cases as language identification is more robust for unique scripts (Kargaran et al., 2023). In these cases, the model often performs worse than random, likely due to not being able to produce responses in the correct format (further details in 5.4).

## 5.3 REGIONAL & ACADEMIC DOMAIN KNOWLEDGE PERFORMANCE

Using the category labels outlined in Section 3.2, we conduct a stratified analysis of five-shot GPT-4o's ability to answer different types of regional questions in **INCLUDE-BASE**. We first note that overall performance differs strongly between languages. In some languages, the model consistently performs well across all academic domains and regional knowledge types, while in others, it struggles across the board (see Appendix Tables 9, 10, and 11).

However, lower performance in certain languages is often linked to questions requiring regional knowledge (*e.g.*, historical knowledge, professional certifications, medical licenses), suggesting that the model's knowledge about regions varies significantly and that its performance across languages reflects this differential. Among regional categories, in Appendix Figure 6, we see the model performs worst on *cultural* questions, followed by *region-explicit* questions. Professional certification exams in different regions are a particular challenge for GPT-4o (average 68.6%). In Persian, the model's accuracy on certification exams is notably low (43.2%), whereas it performs better on subjects such as Geography and Sociology (over 66%). Similarly, in Greek, GPT-4o achieves an average accuracy of 71.3%, but only 54.1% on medical license questions. Further language performance comparisons across various academic disciplines are shown in Appendix Figure 5.

Despite these findings, we note that performance across regional question types (*e.g.*, region-agnostic, cultural) is not deconfounded from other features such as topical difficulty and academic level. Indeed, we observe that *region-agnostic* questions are among the most challenging as models struggle with the mathematical nature of many STEM topics (Frieder et al., 2023; Borges et al., 2024). On average, subjects such as Mathematics and Chemistry show the lowest average accuracies (Appendix Figure 6). Unfortunately, as the region label of any question depends on its subject, it is naturally confounded with the model's ability in that subject, regardless of whether the subject is regional or not.

History is one of the few fields where we can achieve a more controlled study of regional difference as exams can be divided into two categories: those testing region-specific historical knowledge (*e.g.*, "Armenian history", which we label as *cultural*) and general history taught in a particular region (*e.g.*, "World History"; *region-implicit*[5]). In Figure 4, we observe that for all languages that have History exams with both *cultural* and *region-implicit* labels (with the exception of Telugu), the model

---

[4]We denote *intentionally* to mean that this language was reported to be in the pretraining corpus of the model.

[5]We note "World History" as *region-implicit* because the manner in which the subject is taught and evaluated may vary between regions, even if the subject material seems like it should be universal.

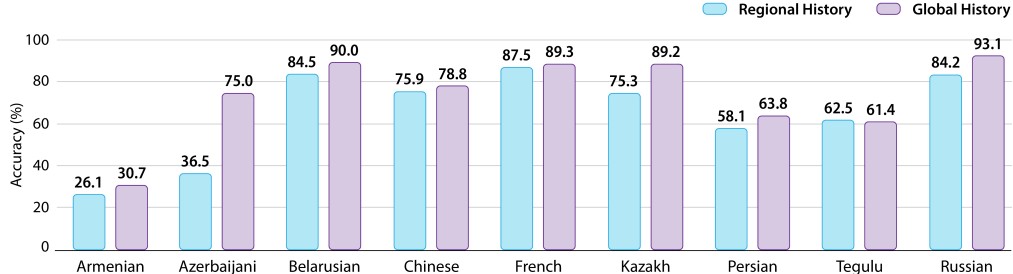

Figure 4: **GPT-4o performance** (In-language Prompt) on regional history exams (*cultural*) and global history exams from that region (*region-implicit*) based on a total of 11,148 questions from INCLUDE. In each language (except Telugu), models perform better on the global history exam.

performs better on the general history exams, indicating a lack of *cultural* knowledge necessary to answer questions for more region-specific topics.

Overall, the variance in performance among different regional categories in our results suggests that model performance on INCLUDE may not be rooted in across-the-board language comprehension issues, but instead in grasping specialized regional knowledge for different languages.

## 5.4 CHALLENGES IN MULTILINGUAL EVALUATION

In our experiments, we observed that models did not always follow the exact format primed by the 5-shot examples or zero-shot instructions (**Format Errors** in Table 3), which required generating a longer output length to rectify.

To empirically measure the impact of this seemingly minute evaluation design choice, we assess the five-shot performance of GPT-4o on INCLUDE-BASE across various decoded output lengths, focusing specifically on its ability to generate a correct response within the first $k$ follow-up tokens ($k$ = 50, 100, 200, and 512). As in our main results, we use the 5-shot prompt template from Hendrycks et al. (2020), without explicitly instructing the model how to generate a correct answers. Instead, the model must induce the format from the provided demonstrations.

Table 12 presents the performance of GPT-4o across the 44 languages of the benchmark, evaluated under four different generation window settings. On average, the model shows a 3.1% performance improvement when increasing the generation length window from 50 to 512 tokens. However, this effect is not uniform; some languages experience significant improvements, such as Uzbek (+17.2%), Armenian (+13.1%), and Malayalam (+12.9%). Many others remain largely unaffected. A manual review and analysis of the generated outputs in languages with the largest gains reveal that the model often generates verbose responses, explaining the context before providing the final answer (*i.e.*, ignoring the formatting in the demonstrations, but reaching the correct response). One possible explanation for these discrepancies is the model's limited ability to leverage in-context learning effectively in certain languages, potentially due to imbalances in language resources during the alignment phase (Zhang et al., 2024c).

When models are prompted with instructions in English, we observe a modest performance improvement of ∼1.5% across all models. Interestingly, in specific cases (*e.g.*, experiments with the Aya-expanse family models, Qwen2.5-14B, and Llama-3.1-Instruct-70B using CoT prompting), we observe significant changes in the frequency of format errors (Table 3). However, this change in frequency does not appear to impact the performance of the models on regional knowledge understanding (Answer Acc., Table 3), suggesting that the choice of instruction language may not significantly enhance or impair the model's ability to reflect regional knowledge, but rather primarily influence the format and consistency of the outputs in certain models.

Overall, these results demonstrate that standardizing evaluation is a challenge in tasks that may lead to different output patterns (Nayab et al., 2024), which is compounded in multilingual evaluations. In particular, given the incentive to lower generation lengths at test time (to lower inference or API costs), reliable multilingual assessment requires anticipating how models will produce outputs in different languages, and how evaluation settings might inadvertently affect measures of performance.

Specifically, practitioners should reflect on penalizing models for format errors when measuring knowledge and intentionally probe for them, given they may not read the languages being evaluated.

## 6 RELATED WORK

In recent years, the creation of benchmarks has substantially improved the evaluation of LLMs. Pioneering efforts such as GLUE and SuperGLUE (Wang et al., 2018; 2019) were important in advancing language understanding tasks. Recent benchmarks, such as MMLU (Hendrycks et al., 2020), HellaSwag (Zellers et al., 2019), ARC (Clark et al., 2018), GSM8K (Cobbe et al., 2021), and BigBench (Srivastava et al., 2022), evaluate models for more complex knowledge comprehension and reasoning. In addition to being final evaluations, they are often used to monitor LLM performance during pretraining, rather than more traditional measures such as perplexity (Penedo et al., 2024). However, these benchmarks use only English data, limiting their utility for multilingual LLMs.

Evaluating multilingual models requires benchmarks that assess models for these same complex abilities across diverse languages. However, multilingual benchmarks initially focused on more basic linguistic abilities (Conneau et al., 2018; Ponti et al., 2020) and collections of such tasks (Liang et al., 2020; Hu et al., 2020; Ruder et al., 2021; Asai et al., 2023; Ahuja et al., 2023a;b). Furthermore, these benchmarks generally include only a few high-resource languages or are based on translations from high-resource languages, limiting the assessment of regional knowledge comprehension and reasoning. Finally, similar to English evaluations, multilingual benchmarks have trended toward saturation (Zhang et al., 2024a; Wang et al., 2024b). Although there have been efforts to create language-specific MMLU-like datasets, coverage remains limited to few languages (Li et al., 2023; Koto et al., 2024; Ghahroodi et al., 2024). Most similar to our proposed effort, the Exams dataset (Hardalov et al., 2020) encompasses questions covering 16 languages collected from elementary and high school science curricula. The Aya dataset (Singh et al., 2024) also includes a sustantial release, covering 513 million data points across 101 languages, including in-language evaluation sets developed by native speakers assessing general performance and safety. However, the Aya dataset is not focused on collecting in-language exams. Our work develops a multilingual benchmark encompassing 44 languages, integrating questions from academic and professional examinations and broadening the evaluation spectrum of multilingual LLMs to include region-specific knowledge.

Finally, a rich body of work has developed benchmarks to assess LLMs for cultural understanding. Arora et al. (2024) evaluate various aspects of culture and language using questions from community forums on 15 topics. Aakanksha et al. (2024) curate a safety dataset that encompasses local nuances. Myung et al. (2024) compile questions about food, sports, holidays, education, and family translated into multiple languages. Synthetic benchmarks, such as NormAd (Rao et al., 2024), generate culturally-rooted stories to measure how well models grasp societal norms. Tools such as CultureBank source cultural descriptions from online platforms such as TikTok (Shi et al., 2024), offering alternative ways to ground cultural benchmarks in dynamic, real-world knowledge. Etxaniz et al. (2024a) analyze the LLM understanding of regional and non-regional knowledge using a parallel evaluation dataset in English and Basque. Beyond benchmarking, Chiu et al. (2024) proposed a tool that facilitates human-machine collaboration for co-creation of complex datasets, challenging the multicultural understanding and adaptability of LLMs. In contrast to this line of work, our study goes beyond culture as a dimension of regional knowledge, and also assesses LLMs on questions that reflect region-related factual knowledge (*e.g.*, professional standards, law, clinical guidelines).

## 7 CONCLUSION

We release **INCLUDE**, a comprehensive multilingual evaluation suite designed to assess performance of large language models (LLMs) across a wide range of subjects and languages for a rich array of cultural and regional knowledge. **INCLUDE** contains 197,243 MCQA pairs from 1,926 examinations across 44 languages and 15 scripts collected from 52 countries. Overall, our results from evaluating 15 models on **INCLUDE** indicate there remains considerable room for model improvement in multilingual regional knowledge understanding and that regional knowledge understanding varies significantly across languages. **INCLUDE** offers researchers and developers a novel and valuable benchmark for evaluating and improving the regional understanding abilities of future multilingual models in the language environments where they would be used.

## ETHICS STATEMENT

The primary goal of our benchmark is to reduce disparities in regional knowledge understanding across languages, addressing the inequities in access to technology and its benefits that often result from these gaps. We have designed the benchmark to reflect a diverse range of linguistic, cultural, and regional contexts, sourcing data from local and region-specific exam materials. Throughout the data collection process, we ensured that no private or sensitive information was included. We only collected data from exams for which there were no license issues. Our benchmark aims to capture and integrate essential cultural knowledge across many languages. We emphasize the importance of local engagement and encourage developers using this benchmark for the evaluation of monolingual models to actively consult with local stakeholders. To promote equitable access to technology and the development of multilingual large language models, we release our benchmark to the community.

**Limitations.** In line with responsible research practices, we acknowledge several limitations in our work. First, INCLUDE spans 44 languages with varying levels of resource availability, leading to different distributions of questions from various academic disciplines across languages. This disparity complicates direct comparisons between performance in disciplines across languages. Additionally, the difficulty of exams may vary not only between languages but also within the same language if exams originate from different sources. However, this limitation is also a reflection of one of the strengths of our benchmark. Questions are sourced from local examinations that reflect the regional and cultural nuances of the environments in which those exams are implemented, which was our motivation for a new evaluation benchmark. Naturally, this precludes exact correspondence between questions across languages. Another practical limitation is that our regional knowledge labels were annotated at the exam topic level, rather than at the individual question level, so questions are classified based on their overarching topic, rather than individual content.

## REPRODUCIBILITY STATEMENT

We release two subsets, INCLUDE-BASE[6] and INCLUDE-LITE[7], alongside the associated documentation and code for data processing and evaluation. These resources will be made publicly available upon acceptance. To mitigate the risk of data contamination during fine-tuning, the full INCLUDE benchmark will be released in incremental stages.

Further details regarding experimental settings, including resource utilization, hyperparameters, and baseline configurations, can be found in Section 4 and Appendix A.5, which provide a comprehensive overview of our methodology.

## ACKNOWLEDGEMENTS

We thank Deniz Bayazit, Badr Alkhamissi, Mete Ismayilzada and Reza Banaei for reading and providing comments on drafts of this paper. We also gratefully acknowledge the support of the Swiss National Science Foundation (No. 215390), Innosuisse (PFFS-21-29), the EPFL Center for Imaging, Sony Group Corporation, and the Allen Institute for AI. This work was supported as part of the Swiss AI Initiative by a grant from the Swiss National Supercomputing Centre (CSCS) under project ID a06 on Alps.

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

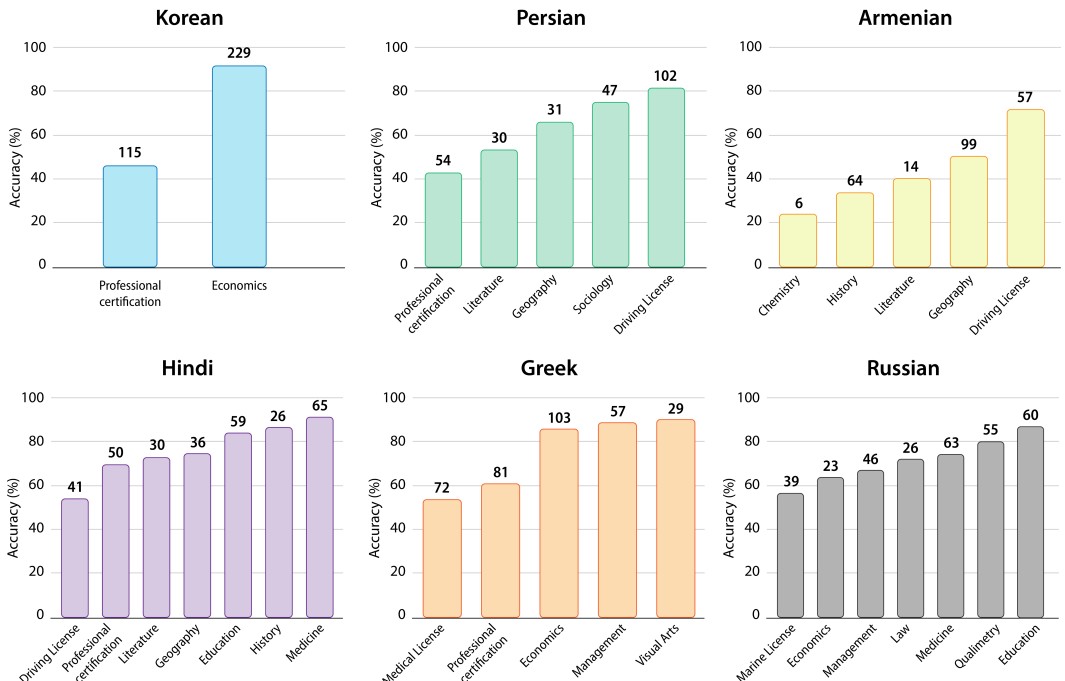

Figure 5: **GPT-4o performance across academic disciplines for Korean, Persian, Armenian, Hindi, Greek, and Russian**. Each bar is annotated with the number of questions with correct answers.

# A APPENDIX

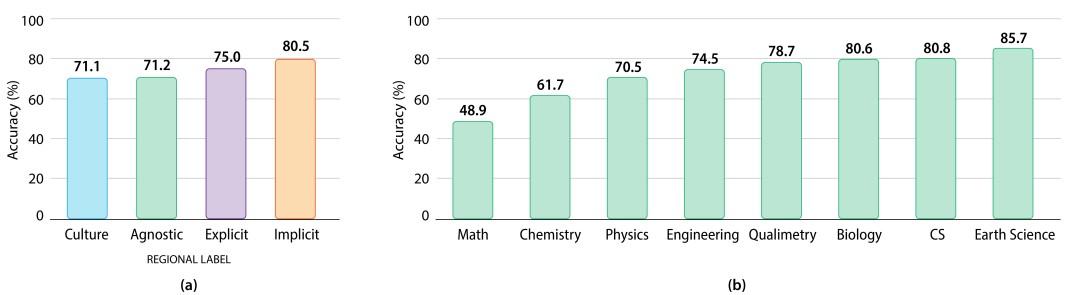

Figure 6: **GPT-4o model performance** on INCLUDE-BASE. (a) Performance across regional labels. While models typically perform better across *region-explicit* and *regional-implicit* questions, it is difficult to disentangle the difficult of questions due to regionality from the subject matter itself (*i.e.*, *region-agnostic* questions may contain more STEM subjects that are traditionally harder for LLMs). (b) Performance across academic disciplines within STEM area. We observe models perform particularly poorly on Math and Chemistry questions.

## A.1 COLLECTED LANGUAGES

Table 8 provides information about the languages in INCLUDE.

| | Accuracy (↑) | | | | |
|---|---|---|---|---|---|
| **Model** | **Humanities** | **STEM** | **Domain-Specific** | **Professional** | **Licenses** |
| # samples | 13294 | 2478 | 1964 | 3165 | 1736 |
| **GPT-4o** | | | | | |
| - 5-shot | 79.0 | 74.2 | 76.8 | 70.1 | 82.1 |
| - Zero-shot CoT | 79.9 | 78.6 | 80.4 | 73.8 | 81.1 |
| **Llama-3.1-70B-Instruct** | | | | | |
| - 5-shot | 71.2 | 69.9 | 74.2 | 64.4 | 73.7 |
| - Zero-shot CoT | 61.9 | 57.5 | 63.5 | 56.7 | 58.4 |
| **Aya-expanse-32B** | | | | | |
| - 5-shot | 49.6 | 43.0 | 49.1 | 34.7 | 49.5 |
| - Zero-shot CoT | 52.9 | 47.8 | 55.4 | 44.3 | 52.9 |
| **Qwen2.5-14B** | | | | | |
| - 5-shot | 61.4 | 60.9 | 66.0 | 57.1 | 65.1 |
| - Zero-shot CoT | 48.6 | 44.4 | 51.6 | 41.6 | 46.9 |
| **Aya-expanse-8B** | 37.8 | 32.3 | 37.3 | 40.2 | 29.7 |
| **Mistral-7B (v0.3)** | 44.2 | 43.4 | 43.9 | 38.6 | 44.3 |
| **Mistral-7B-Instruct (v0.3)** | 44.5 | 42.7 | 43.2 | 40.1 | 43.7 |
| **Gemma-7B** | 55.1 | 53.6 | 55.5 | 47.7 | 62.2 |
| **Gemma-7B-Instruct** | 38.6 | 37.7 | 42.0 | 34.5 | 44.9 |
| **Qwen2.5-7B** | 53.4 | 54.2 | 59.1 | 51.3 | 57.8 |
| **Qwen2.5-7B-Instruct** | 53.5 | 53.3 | 58.1 | 49.5 | 58.6 |
| **Llama-3-8B** | 51.7 | 49.8 | 52.1 | 43.4 | 51.3 |
| **Llama-3-8B-Instruct** | 50.7 | 46.9 | 52.9 | 44.3 | 54.4 |

Table 2: Accuracy performance of GPT-4o (In-language prompting) on INCLUDE-BASE grouped by high-level topics. **Humanities** include Social Science, Humanities, and General knowledge. **STEM** includes Applied Science and STEM. **Domain-specific** covers Business & Commerce and Health oriented education. **Professional** includes professional certifications. **Licenses** cover Marine, Fishing, and Driving licenses.

| | In-language Prompt | | | English Prompt | | |
|---|---|---|---|---|---|---|
| **Model** | **Total Acc.** | **Answer Acc.** | **Format Errors (%)** | **Total Acc.** | **Answer Acc.** | **Format Errors (%)** |
| **GPT-4o** | | | | | | |
| - 5-shot | 77.3 | 79.0 | 2.5 | 76.3 | 78.0 | 2.2 |
| - Zero-shot CoT | 79.0 | 79.2 | 0.2 | 78.9 | 79.1 | 0.2 |
| **Llama-3.1-70B-Instruct** | | | | | | |
| - 5-shot | 70.6 | 70.6 | 0.0 | 70.7 | 70.7 | 0.0 |
| - Zero-shot CoT | 60.6 | 67.9 | 10.9 | 56.3 | 67.8 | 17.0 |
| **Aya-expanse-32B** | | | | | | |
| - 5-shot | 52.4 | 56.2 | 16.9 | 56.6 | 62.7 | 9.7 |
| - Zero-shot CoT | 51.4 | 57.2 | 10.2 | 57.7 | 58.4 | 1.1 |
| **Qwen2.5-14B** | | | | | | |
| - 5-shot | 61.4 | 62.4 | 1.5 | 61.7 | 61.7 | 0.0 |
| - Zero-shot CoT | 47.3 | 53.1 | 10.9 | 51.0 | 52.0 | 1.9 |
| **Aya-expanse-8B** | 37.2 | 43.8 | 18.0 | 46.0 | 50.7 | 9.2 |
| **Mistral-7B (v0.3)** | 43.3 | 43.3 | 0.0 | 44.9 | 44.9 | 0.0 |
| **Mistral-7B-Instruct (v0.3)** | 43.6 | 43.8 | 0.4 | 44.5 | 44.5 | 0.1 |
| **Gemma-7B** | 54.5 | 54.5 | 0.0 | 54.9 | 54.9 | 0.0 |
| **Gemma-7B-Instruct** | 38.7 | 38.7 | 0.0 | 39.7 | 39.7 | 0.1 |
| **Qwen2.5-7B** | 54.1 | 55.1 | 1.9 | 55.2 | 55.2 | 0.0 |
| **Qwen2.5-7B-Instruct** | 53.8 | 54.0 | 0.5 | 54.6 | 54.6 | 0.0 |
| **Llama-3.1-8B** | 51.0 | 51.0 | 0.0 | 51.8 | 51.8 | 0.0 |
| **Llama-3.1-8B-Instruct** | 53.4 | 53.4 | 0.0 | 54.6 | 54.6 | 0.0 |

Table 3: Results on INCLUDE-BASE for *In-language* and *English* prompting strategies. **Total Accuracy** represents the raw accuracy of the model for answering INCLUDE questions in each respective subset. **Answer Accuracy** represents the accuracy of the model when only considering samples where an answer is extracted from the model's output in the correct response format. **Formatting Errors (%)** describes the percentage of model responses that are not formatted correctly and so do not output any answer option. We mark these incorrect by default in **Total Accuracy** and do not include them when computing **Answer Accuracy**.

| Model | INCLUDE-LITE | | INCLUDE-BASE | |
|---|---|---|---|---|
| | In-Language Prompt | English Prompt | In-Language Prompt | English Prompt |
| **Llama3.1-70B-Instruct** | 70.3 | 70.6 | 70.6 | 70.9 |
| **Aya-expanse-32B** | 58.9 | 59.5 | 47.2 | 47.8 |
| **Qwen2.5-14B** | 61.8 | 61.9 | 62.3 | 62.6 |
| **Aya-expanse-8B** | 47.3 | 48.0 | 47.2 | 47.8 |
| **Mistral-7B** | 44.5 | 44.7 | 44.1 | 44.6 |
| **Mistral-7B-Instruct** | 43.8 | 43.9 | 44.2 | 44.3 |
| **Gemma-7B** | 53.6 | 53.1 | 53.5 | 53.2 |
| **Gemma-7B-Instruct** | 39.1 | 39.7 | 38.6 | 39.3 |
| **Qwen2.5-7B** | 54.4 | 54.9 | 55.0 | 55.5 |
| **Qwen2.5-7B-Instruct** | 54.5 | 54.6 | 54.8 | 54.8 |
| **Llama-3.1-8B** | 51.2 | 52.1 | 51.2 | 51.9 |
| **Llama-3.1-8B-Instruct** | 53.5 | 54.4 | 53.5 | 54.4 |

Table 4: Harness evaluation results on INCLUDE-BASE.

| Academic area | Academic field | Label |
|---|---|---|
| Humanities | Logic | Agnostic |
| | Law | Region Explicit |
| | Language | Culture |
| | Visual Arts, History, Philosophy, Religious studies, Performing arts, Culturology, Literature | Region implicit/ Culture |
| Social Science | Sociology, Political sciences, Anthropology | Region implicit/Culture |
| | Economics | Region implicit/Agnostic/Region explicit |
| | Psychology | Region implicit/Region explicit |
| | Geography | Region implicit/Agnostic |
| STEM | Math, Physics, CS, Biology, Earth science, Chemistry, Engineering | Agnostic |
| | Qualimetry | Region explicit |
| Health oriented education | Medicine | Agnostic/Region implicit/Region explicit |
| | Health | Region implicit/Region explicit |
| Business and Commerce | Accounting | Region explicit |
| | Management, Marketing, Industrial and labor relations, International trade, Risk management and insurance, Business administration, Business ethics, Business, Finance | Region implicit/Region explicit/Agnostic |
| Applied Science | Agriculture, Library and museum studies, Transportation | Region implicit/Agnostic |
| | Military Sciences, Public Administration, Public Policy | Region implicit/Region explicit |
| | Architecture and Design, Family and consumer science, Environmental studies and forestry, Education Journalism, media studies, and communication, Social Work, Human physical performance and recreation | Region implicit |
| Other | Driving license, Marine license, Fishing license, Medical license, Public administration, Professional certification | Region explicit |
| General knowledge | Multiple exams | Region implicit/Culture |

Table 5: Annotation schema for high-level **Academic area** and fine-grained **Academic field**. The **Label** column lists the most likely *regionality* label for these exams in our dataset (*e.g.*, region-{*agnostic, implicit, explicit*} or *cultural*), though all exams from which we collect data are individually labeled with a *regionality* category. The first label is the most frequent one.

| Language | Academic Humanities | Academic STEM studies | Academic Domain-specific studies | Professional | License | Avg (%) |
|---|---|---|---|---|---|---|
| **Albanian** | 95.0 | 88.0 | 83.5 | - | - | 89.50 |
| **Arabic** | 77.8 | 82.0 | 80.5 | - | 76.2 | 78.30 |
| **Armenian** | 52.7 | 32.0 | - | - | 72.2 | 53.60 |
| **Azerbaijani** | 71.3 | 73.6 | 71.4 | - | - | 71.90 |
| **Basque** | - | - | - | 64.8 | - | 64.80 |
| **Belarusian** | 51.8 | 42.0 | - | - | - | 50.90 |
| **Bengali** | 71.1 | 90.0 | - | 84.3 | - | 76.80 |
| **Bulgarian** | 93.8 | 60.0 | - | - | - | 90.70 |
| **Chinese** | 71.5 | 66.7 | 58.2 | 52.1 | 84.5 | 66.10 |
| **Croatian** | 89.0 | 82.0 | - | - | - | 88.40 |
| **Dutch; Flemish** | 86.6 | 87.5 | 80.0 | - | - | 86.40 |
| **Estonian** | 90.7 | 98.0 | 100.0 | - | - | 92.40 |
| **Finnish** | 67.0 | 87.0 | 77.8 | - | - | 69.90 |
| **French** | 83.8 | 50.0 | 81.2 | - | 68.1 | 80.70 |
| **Georgian** | 87.6 | - | - | - | - | 87.60 |
| **German** | 62.6 | 64.0 | - | - | 87.0 | 66.90 |
| **Greek** | 84.7 | 84.0 | 89.2 | 58.6 | - | 71.50 |
| **Hebrew** | 62.0 | - | - | - | 88.6 | 86.20 |
| **Hindi** | 77.7 | 71.9 | 91.5 | 71.8 | 57.7 | 75.10 |
| **Hungarian** | 66.3 | 80.6 | - | - | - | 75.80 |
| **Indonesian** | 84.0 | 69.1 | - | 84.8 | - | 79.50 |
| **Italian** | 87.7 | 87.2 | 91.7 | 95.5 | - | 90.00 |
| **Japanese** | - | - | - | 78.1 | 96.0 | 81.60 |
| **Kazakh** | 80.4 | - | - | - | - | 80.40 |
| **Korean** | 91.6 | - | - | 46.4 | - | 69.00 |
| **Lithuanian** | 92.0 | 97.1 | 82.5 | 81.2 | - | 90.60 |
| **Malay** | 84.5 | - | 80.3 | - | - | 83.00 |
| **Malayalam** | 69.6 | 66.0 | 55.0 | - | 80.9 | 70.80 |
| **Nepali** | - | - | - | 61.6 | 83.2 | 72.40 |
| **Macedonian** | 96.0 | 86.0 | 89.3 | - | - | 92.40 |
| **Persian** | 66.0 | 25.0 | - | 49.6 | 81.6 | 64.60 |
| **Polish** | 100.0 | 64.6 | - | 80.0 | - | 78.80 |
| **Portuguese** | 84.7 | 63.3 | 67.9 | - | - | 76.40 |
| **Serbian** | 92.2 | 86.0 | - | - | - | 91.60 |
| **Spanish** | 83.6 | 88.0 | 96.0 | - | - | 84.40 |
| **Tagalog** | 86.8 | - | - | - | 90.7 | 87.40 |
| **Tamil** | 70.6 | 54.0 | - | - | - | 69.10 |
| **Telugu** | 66.9 | 70.7 | - | - | - | 68.20 |
| **Turkish** | 62.0 | 52.0 | 75.9 | - | - | 65.30 |
| **Ukrainian** | 85.8 | 84.0 | - | - | - | 85.60 |
| **Urdu** | 61.7 | 65.3 | 100.0 | - | - | 62.50 |
| **Uzbek** | 63.6 | 84.0 | - | 73.3 | - | 69.70 |
| **Vietnamese** | 84.4 | 86.0 | - | - | - | 84.50 |
| **Russian** | 77.5 | 83.4 | 70.8 | - | 63.9 | 75.00 |

Table 6: Accuracy performance of GPT-4o (5-shot) on INCLUDE-BASE for each language. **Humanities** include Social Science, Humanities, and General knowledge. **STEM** includes Applied Science and STEM. **Domain-specific** covers Business & Commerce and Health oriented education. **Professional** includes professional certifications. **Licenses** cover Marine, Fishing, and Driving licenses.

| | Aya-expanse-8B | XGLM-7B | Qwen-2.5-7B | LLaMA-3.1-8B |
|---|---|---|---|---|
| **Full Benchmark** | 0.02 | 0.17 | 0.13 | 0.29 |
| **Newly collected** | 0.01 | 0.14 | 0.11 | 0.25 |

Table 7: Data contamination rates per model on INCLUDE-BASE.

| Language | Script | Family | Branch | Availability | Count |
|---|---|---|---|---|---|
| **Albanian** | latin | Indo-European | Albanian | Mid | 2365 |
| **Amharic** | ge'ez | Afro-Asiatic | Semitic | Low | 131 |
| **Arabic** | perso-arabic | Afro-Asiatic | Semitic | High | 15137 |
| **Armenian** | armenian | Indo-European | Armenian | Low | 1669 |
| **Assamese** | bengali-assamese | Indo-European | Indo-Iranian | Low | 323 |
| **Azerbaijani** | latin | Turkic | Azerbaijani North | Mid | 6937 |
| **Basque** | latin | Isolate | | Low | 719 |
| **Belarusian** | cyrillic | Indo-European | Slavic East | Low | 687 |
| **Bengali** | bengali-assamese | Indo-European | Indo-Iranian | Mid | 15259 |
| **Bulgarian** | cyrillic | Indo-European | Slavic South Eastern | Mid | 2937 |
| **Chinese** | chinese | Sino-Tibetan | Chinese | High | 12977 |
| **Croatian** | latin | Indo-European | Slavic South Western | Mid | 2879 |
| **Czech** | latin | Indo-European | Slavic West | High | 50 |
| **Danish** | latin | Indo-European | Germanic | Mid | 732 |
| **Dutch; Flemish** | latin | Indo-European | Germanic | High | 2222 |
| **Estonian** | latin | Uralic | Finnic | Mid | 952 |
| **Finnish** | latin | Uralic | Finnic | Mid | 1574 |
| **French** | latin | Indo-European | Italic | High | 2457 |
| **Georgian** | mkherduli | Kartvelian | Georgian | Low | 599 |
| **German** | latin | Indo-European | Germanic | High | 1590 |
| **Greek** | greek | Indo-European | Greek | Mid | 6570 |
| **Hebrew** | hebrew | Afro-Asiatic | Semitic | Mid | 2457 |
| **Hindi** | devanagari | Indo-European | Indo-Iranian | Mid | 5167 |
| **Hungarian** | latin | Uralic | Hungarian | Mid | 2267 |
| **Indonesian** | latin | Austronesian | Malayo-Polynesian | High | 12013 |
| **Italian** | latin | Indo-European | Italic | High | 3038 |
| **Japanese** | kanji | Japonic | Japanese | High | 2699 |
| **Kannada** | kannada | Dravidian | Southern | Low | 335 |
| **Kazakh** | cyrillic | Turkic | Western | Low | 5736 |
| **Korean** | hangul | Koreanic | Korean | Mid | 1781 |
| **Lithuanian** | latin | Indo-European | Eastern Baltic | Mid | 1397 |
| **Malay** | latin | Austronesian | Malayo-Polynesian | Mid | 1021 |
| **Malayalam** | vatteluttu | Dravidian | Southern | Low | 275 |
| **Marathi** | devanagari | Indo-European | Indo-Iranian | Mid | 313 |
| **Nepali** | devanagari | Indo-European | Indo-Iranian | Mid | 1470 |
| **Macedonian** | cyrillic | Indo-European | Slavic South Eastern | Low | 2075 |
| **Oriya** | odia | Indo-European | Indo-Iranian | Low | 241 |
| **Panjabi; Punjabi** | gurmukhi | Indo-European | Indo-Iranian | Low | 453 |
| **Persian** | perso-arabic | Indo-European | Indo-Iranian | High | 23990 |
| **Polish** | latin | Indo-European | Slavic West | High | 2023 |
| **Portuguese** | latin | Indo-European | Italic | High | 1407 |
| **Russian** | cyrillic | Indo-European | Slavic East | High | 10169 |
| **Serbian** | cyrillic | Indo-European | Slavic South | Mid | 1636 |
| **Sinhala; Sinhalese** | sinhala | Indo-European | Indo-Iranian | Low | 325 |
| **Slovak** | latin | Indo-European | Slavic West | Mid | 131 |
| **Spanish** | latin | Indo-European | Italic | High | 2559 |
| **Swedish** | latin | Indo-European | Germanic | Mid | 5102 |
| **Tagalog** | latin | Austronesian | Malayo-Polynesian | Low | 530 |
| **Tamil** | tamil | Dravidian | Southern | Mid | 945 |
| **Telugu** | telugu | Dravidian | South-Central | Low | 11568 |
| **Turkish** | latin | Turkic | Southern | High | 2710 |
| **Ukrainian** | cyrillic | Indo-European | Slavic East | Mid | 1482 |
| **Urdu** | perso-arabic | Indo-European | Indo-Iranian | Low | 122 |
| **Uzbek** | latin | Turkic | Eastern | Low | 2878 |
| **Vietnamese** | latin | Austro-Asiatic | Mon-Khmer | High | 8901 |

Table 8: Languages in **INCLUDE** with their associated metadata and the total count of the samples per language.

## A.2 DETAILS ON PARSING EXAM SOURCES

Figure 8 presents the questionnaire we distributed to the community to gather a diverse set of multiple-choice exams. It was distributed among university student organizations and researchers at our institution.[8] Participation was voluntary and not incentivized.

## A.3 PERFORMANCE ACROSS ACADEMIC AREAS AND FIELDS

Distribution of academic areas and academic fields with the respective number of questions is presented in Figure 7. GPT-4o performance across languages and academic areas is in Table 9. GPT-4o performance across languages, academic fields, and related regional features is in Tables 10 and 11.

## A.4 REGIONAL LABELS: ANNOTATION

First, we categorized the exams into one of eight broad academic areas, *e.g.*, Humanities or Social Sciences, and then further classified each exam into a specific academic fields, *e.g.*, History or Geography. This categorization was done manually, taking into account both the exam's learning level and the exam's original topic.

Building on these categories, we applied one of four labels—agnostic, culture-related, region-explicit, or region-implicit—based on the degree of dependence on localized knowledge required to answer the exam questions. The labels reflect the extent to which specific cultural or regional knowledge is necessary. Table 5 provides examples illustrating how different exams were typically classified under each label, to show the relationship between categories and labels.

The "region implicit" label was applied when we suspected that exam content might vary across regions but could not reliably detect specific regional differences. For example, historical events, literary works, and religious interpretations may differ significantly depending on the region. Similarly, fields like marketing, management, social work, and insurance—though rooted in shared theoretical foundations—can be practiced differently across regions. When we encountered such uncertainty, we labeled the subject as "region implicit."

Within the Humanities, fields such as Visual Arts, History, Philosophy, Religious Studies, Performing Arts, Culturology, and Literature were labeled "region implicit" when the content was not explicitly tied to a particular region. However, if the exam was region-specific (*e.g.*, Greek literature), we categorized it as "culture-related."

In the Social Sciences, Psychology was classified as "region explicit" if the exam focused on regional clinical practices; otherwise, it was "region implicit" when dealing with broader psychological theories that may vary across regions. Geography was labeled "region implicit" if the exam involved political geography and "agnostic" if it focused on general geographic knowledge. Similarly, disciplines like Sociology, Political Science, and Anthropology were classified as either "region implicit" or "culture-related," depending on whether regional specificity was required, much like History.

For Economics, exams were labeled "agnostic" when covering general economic theories, "region explicit" when addressing regional regulations, and "region implicit" when regional applications were uncertain. In STEM fields, most disciplines were categorized as "agnostic," with the exception of Qualimetry, which was labeled "region explicit" due to its specific application in post-Soviet countries for quantitative and qualitative assessment according to regional standards.[9]

Exams related to theoretical medical subjects, such as Anatomy, were classified as "agnostic." In contrast, exams covering clinical practices and guidelines specific to a region were labeled as "region explicit," while others were marked as "region implicit" if regional dependence was unclear.

Accounting is generally tied to region-specific practices, so it was consistently classified as "region explicit." Other disciplines within the Business and Commerce category were treated similarly to Economics and labeled as mostly as "region implicit." In some cases, there were "agnostic" and "region explicit" exams.

---

[8]We will provide institutional details upon paper publication.
[9]https://en.wikipedia.org/wiki/Qualimetry

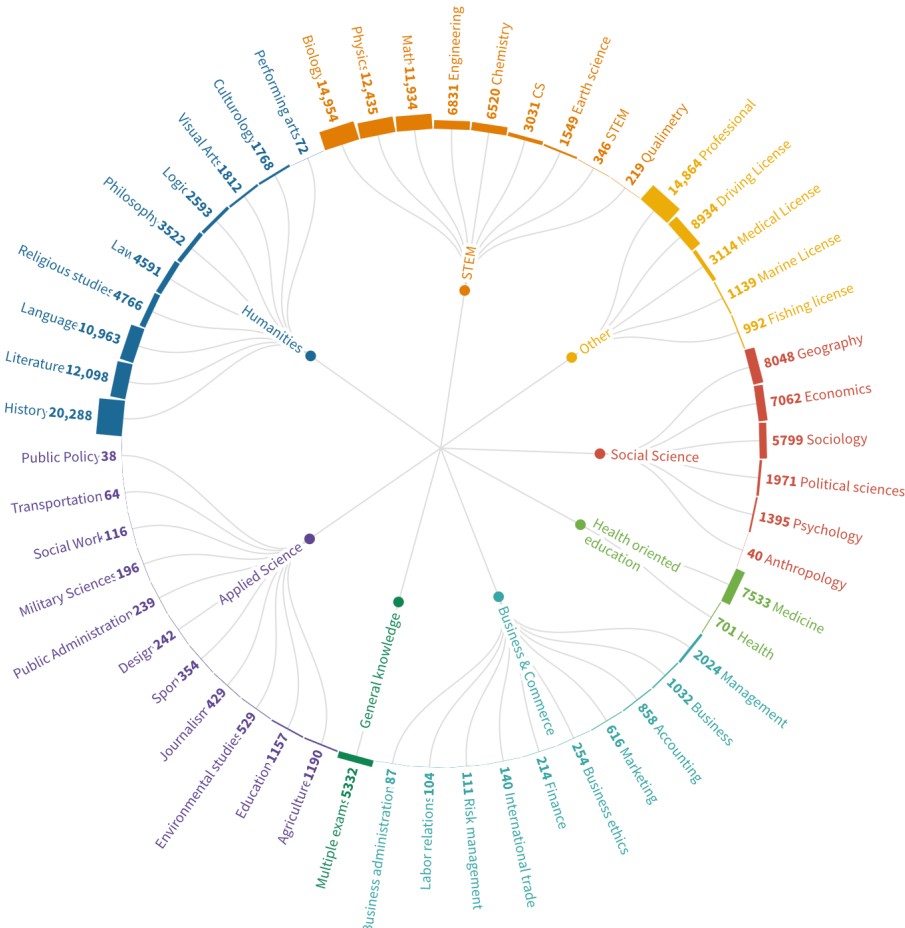

Figure 7: **Academic domain and academic fields with the number of examples across all languages.**

Exams in Applied Science disciplines were typically categorized as "region implicit" due to the potential involvement of regional variations. Similarly, exams in Military Sciences, Public Administration, and Public Policy were marked as "region explicit" when tied to specific regions (*e.g.*, Basics of National Security of the Republic of Azerbaijan) and "region implicit" when regional specifics were less pronounced. For exams focused on theoretical aspects, we used the "agnostic" label (*e.g.*, Theoretical Foundations of Food Engineering in Agriculture).

Finally, exams covering multiple topics were classified as "region implicit" unless they explicitly focused on cultural aspects of a particular region, in which case they were labeled as "culture-related."

## A.5 IMPLEMENTATION DETAILS

Each model was evaluated using a single A100 GPU (80GB memory), with evaluation times averaging approximately 4 hours for **INCLUDE-BASE**. For all models, we set the decoding temperature to 0, prioritizing deterministic outputs.

We configured response context windows based on model size and task requirements. For models such as Aya-23-8B, Mistral-7B (v0.3), Mistral-7B-Instruct (v0.3), Gemma-7B, Gemma-7B-Instruct, Qwen2.5-7B, Qwen2.5-7B-Instruct, Llama-3-8B, Llama-3-8B-Instruct, XGLM-7.5B, BLOOM-7.1B,

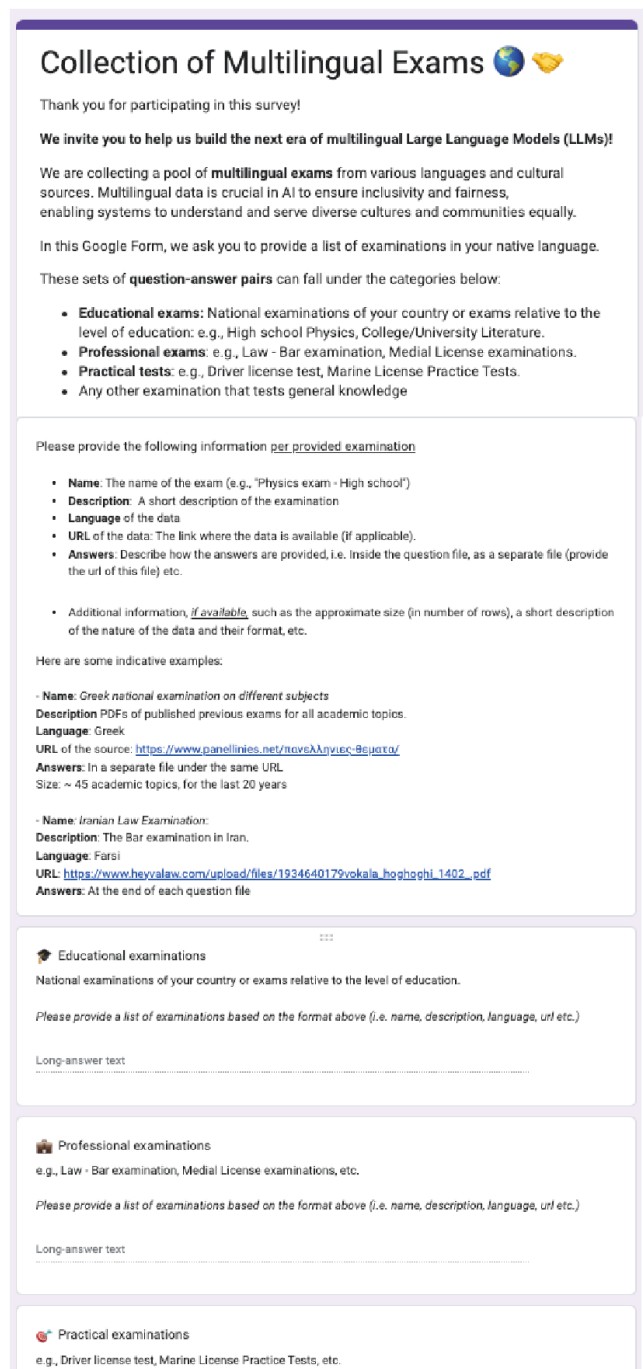

Figure 8: Exam source collection form sent to the academic community.

and BLOOMZ-7.1B, we set a window size of 40 tokens. Larger models, including C4AI-Aya-23-35B and GPT-4, were evaluated using a 512-token context window for 5-shot tasks and 1024 tokens for zero-shot chain-of-thought (CoT) reasoning.

## A.6 EXPERIMENTS ON MONOLINGUAL MODELS

To assess the performance of monolingual models on our benchmark, we evaluate seven open-source monolingual models on the relevant language-specific subsets of INCLUDE, (i.e., the languages these

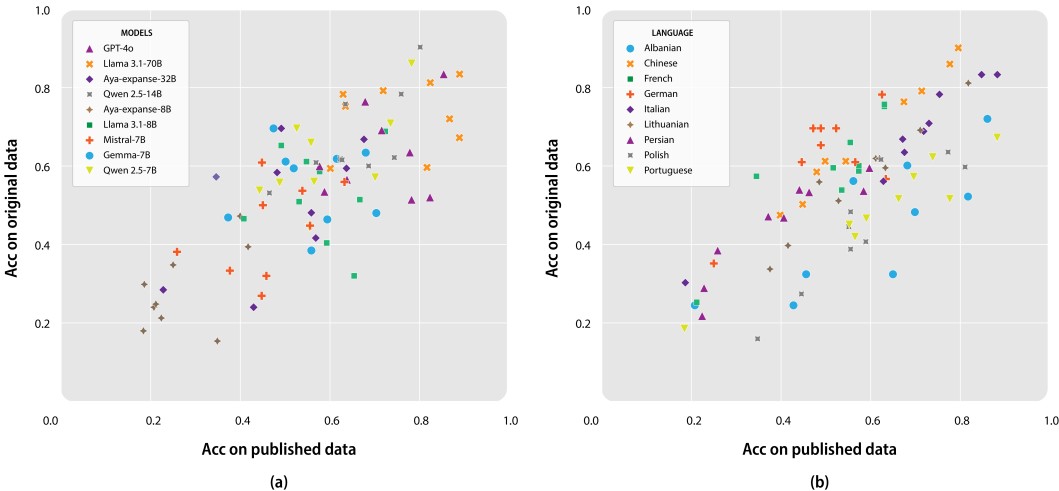

Figure 9: Accuracy of different models on languages where both existing benchmark data and newly collected data are available. Each point represents the accuracy score of a model for a specific language. (a) Points of the same color represent the accuracy scores of a single model across different languages. (b) Points of the same color represent the accuracy scores for a single language across different models.

models were pre-trained on). We test Baichuan-7B (Yang et al., 2023), SILMA-9B-Instruct (Team, 2024), Calm2-7B-chat (Touvron et al., 2023), Korean-Mistral, ruGPT3.5-13B and SauerkrautLM-v2-14b-DPO. We compare their performance with the results of the most performant large-, medium-, and small-scale models on the specific language subsets. The results of this evaluation are presented in the Table 13. The table reveals that most monolingual models underperform the state-of-the-art small multilingual model Qwen-2.5 (7B), with the exception of the German monolingual model, SauerkrautLM-v2-14B-DPO, which performs on par with Qwen in the German language.

## A.7 LIMITATIONS OF THE SCOPE OF EXISTING BENCHMARKS

As discussed in the motivation and related work sections, there is currently no multilingual benchmark that offers both high language coverage and incorporates regional knowledge. Existing benchmarks typically fall into one of two categories: either they focus extensively on a single language across various regional dimensions, or they cover a certain number of languages with predominantly *region-agnostic* knowledge (*e.g.*, STEM). In Table 15, we provide more details on the list of existing benchmarks that were mentioned in the paper, which feature original content (not machine-translated), highlighting their language and knowledge coverage.

In relation to these monolingual existing resources, the **INCLUDE** benchmark makes three significant contributions: (1) introduces original datasets from languages that are either not covered or only partially covered by existing benchmarks, (2) leverages publicly available, knowledge-intensive multiple-choice question benchmarks in various languages, (3) organizes both the existing and newly introduced data under a unified taxonomy of knowledge, differentiating between regional knowledge and region-agnostic knowledge. In this context, **INCLUDE** incorporates existing datasets, which account for 39.8% of the total collected data and 31.7% of the **INCLUDE-BASE** benchmark. To understand the correlations between the newly collected data and the existing benchmarks integrated into **INCLUDE**, we analyzed model performance in languages where both existing benchmark data and newly collected data are available. We compared performance across these datasets and examined the correlation between them. Results were stratified by language and by model type. We visualized the performances using plots (Figure 9) and calculated the $R^2$ scores to quantify the correlations (Table 14).

The analysis reveals two key conclusions: First, for a given language with multiple models having published performance data, a model's performance on **INCLUDE** can generally be predicted. However, for a given model with published performance across different languages, its performance on

INCLUDE cannot reliably be predicted when applied to a newly published language benchmark. This indicates that while INCLUDE is less impactful for languages with existing published benchmarks, it is particularly valuable for assessing performance in languages with no prior resources.

## A.8 ANALYSIS OF OUTPUT ERRORS ON INCLUDE

As outlined in Section 5.3 of the paper, model performance—and consequently, the errors—are heavily influenced by the model's proficiency in the specific task and language. We conducted a more detailed error analysis by manually investigating a sample of INCLUDE generations. We focused on six languages spanning high-resource (Chinese, Turkish), medium-resource (Bengali, Greek, Korean), and low-resource (Armenian) categories, selecting subject areas with the largest performance gaps compared to other subjects within the same language. We manually examined 150 examples, covering at least two subjects per language with 10 examples per subject, analyzing the questions and answers generated by GPT-4o. We observed four main types of errors related to computational mistakes, factual mistakes, lack of regional knowledge, and format errors (Table 16). Each error type highlights distinct limitations in the model's capabilities, from arithmetic and factual knowledge to regional understanding and prompt adherence. Our analysis revealed that the model's errors were distributed as follows: 38.6% were due to a lack of regional knowledge, 32% resulted from format errors, 26.7% were factual mistakes, and 2.7% were computational errors.

## A.9 DATA CONTAMINATION PREVENTION

As described in Section 3.1, INCLUDE is made up of a few previously-published benchmarks incorporated into INCLUDE, but also newly-collected exam materials from sources contributed by our multilingual community of native speakers. A significant portion of the newly-collected data was derived from PDFs and textbooks, which are less likely to have been included in models trained primarily on web-based data.

In this section, we analyze the degree of contamination within the models using the mink%++ (Zhang et al., 2024b) method for training data detection in LLMs. This method determines whether an input next token forms a mode or has a relatively high probability under the conditional categorical distribution. Using this scoring mechanism, one can predict if an input sequence is part of the model's training data based on a decision threshold. This method achieves SOTA on the WikiMIA (Shi et al., 2023) benchmark for training data detection. We use the decision threshold that achieves the best performance on WikiMIA as the decision threshold for our analysis. Using this method, we computed the contamination rate for each language on four main-stream multilingual models: Aya-8B, XGLM-7B, LLaMA-3.1-8B, and Qwen-2.5-7B. We show the contamination rate results in Table 7.

To further mitigate the risk of benchmark saturation as a result of data leakage when new models are trained, we have held back the complete dataset, comprising 197,243 entries. Instead, we will release these further questions and answers incrementally over the next year. We have also reserved a held-out dataset covering a wide range of the collected languages to be used for future experimental studies specifically aimed at analyzing data leakage over time.

| Language | Academic Area | Accuracy | Count |
|---|---|---|---|
| Albanian | Humanities | 95.1 | 223 |
| | Business & Commerce | 85.7 | 223 |
| | Social Science | 94.5 | 55 |
| Arabic | Humanities | 79.0 | 105 |
| | Business & Commerce | 79.3 | 82 |
| | General Knowledge | 86.7 | 105 |
| | Other | 76.2 | 105 |
| | STEM | 82.0 | 50 |
| | Social Science | 67.6 | 105 |
| Armenian | Humanities | 34.7 | 225 |
| | Other | 72.2 | 79 |
| | STEM | 28.0 | 50 |
| | Social Science | 50.5 | 196 |
| Azerbaijani | Applied Science | 75.9 | 108 |
| | Humanities | 74.1 | 108 |
| | Business & Commerce | 62.5 | 96 |
| | Health-Oriented Education | 80.2 | 96 |
| | Social Science | 67.6 | 108 |
| Basque | Other | 64.8 | 500 |
| Belarusian | Humanities | 50.8 | 490 |
| | STEM | 42.0 | 50 |
| Bengali | Humanities | 62.0 | 166 |
| | General Knowledge | 80.1 | 166 |
| | Other | 84.3 | 166 |
| | STEM | 88.0 | 50 |
| Bulgarian | Humanities | 96.4 | 250 |
| | STEM | 60.0 | 50 |
| | Social Science | 91.2 | 250 |
| Chinese | Applied Science | 73.2 | 71 |
| | Humanities | 67.8 | 87 |
| | Business & Commerce | 53.5 | 71 |
| | Health-Oriented Education | 60.9 | 87 |
| | Other | 68.3 | 142 |
| | Social Science | 76.1 | 71 |
| Croatian | Humanities | 86.8 | 250 |
| | STEM | 82.0 | 50 |
| | Social Science | 90.8 | 250 |
| Dutch; Flemish | Humanities | 86.0 | 243 |
| | Social Science | 86.8 | 243 |
| Estonian | Humanities | 90.1 | 161 |
| | STEM | 97.2 | 36 |
| Finnish | Humanities | 69.5 | 226 |
| | Health-Oriented Education | 75.6 | 45 |
| | Social Science | 64.6 | 226 |
| French | Humanities | 86.5 | 266 |
| | Other | 68.1 | 47 |
| | Social Science | 74.3 | 74 |
| Georgian | Humanities | 87.6 | 500 |
| German | Social Science | 62.6 | 91 |
| Greek | Humanities | 83.8 | 37 |
| | Business & Commerce | 89.1 | 64 |
| | Other | 57.5 | 266 |
| | Social Science | 84.2 | 133 |
| Hebrew | Humanities | 60.0 | 50 |
| | Other | 88.6 | 500 |
| Hindi | Applied Science | 83.1 | 71 |
| | Humanities | 72.9 | 96 |
| | General Knowledge | 83.1 | 71 |
| | Health-Oriented Education | 91.5 | 71 |
| | Other | 64.1 | 142 |
| | Social Science | 74.6 | 71 |
| Hungarian | Applied Science | 79.8 | 341 |
| | Social Science | 66.3 | 184 |
| Indonesian | Applied Science | 71.2 | 125 |
| | Humanities | 82.4 | 125 |
| | Other | 83.2 | 125 |
| | STEM | 60.0 | 50 |
| | Social Science | 84.8 | 125 |
| Italian | Applied Science | 85.7 | 35 |
| | Humanities | 85.0 | 167 |
| | Other | 95.5 | 155 |
| | Social Science | 89.8 | 167 |

| Language | Academic Area | Accuracy | Count |
|---|---|---|---|
| Japanese | Other | 80.2 | 501 |
| Kazakh | Humanities | 80.4 | 500 |
| Korean | Other | 46.0 | 250 |
| | Social Science | 91.6 | 250 |
| Lithuanian | Humanities | 91.6 | 335 |
| | Business & Commerce | 77.5 | 40 |
| | Other | 81.2 | 48 |
| | STEM | 97.1 | 34 |
| | Social Science | 93.5 | 77 |
| Malay | Humanities | 84.3 | 178 |
| | Business & Commerce | 79.8 | 178 |
| | Social Science | 84.8 | 145 |
| Malayalam | Humanities | 64.3 | 56 |
| | General Knowledge | 73.1 | 78 |
| | Health-Oriented Education | 55.0 | 100 |
| | Other | 80.9 | 194 |
| | STEM | 66.0 | 47 |
| Nepali | Other | 72.4 | 500 |
| Macedonian | Humanities | 96.9 | 224 |
| | Business & Commerce | 89.3 | 224 |
| | STEM | 86.0 | 50 |
| | Social Science | 92.5 | 53 |
| Persian | Humanities | 55.3 | 141 |
| | Other | 62.4 | 250 |
| | Social Science | 74.5 | 141 |
| Polish | Other | 80.0 | 496 |
| | STEM | 62.5 | 48 |
| Portuguese | Applied Science | 58.3 | 84 |
| | Humanities | 81.8 | 154 |
| | Business & Commerce | 56.9 | 84 |
| | Health-Oriented Education | 67.1 | 67 |
| | Other | 67.6 | 169 |
| Russian | Applied Science | 87.0 | 69 |
| | Humanities | 76.8 | 69 |
| | Business & Commerce | 66.7 | 69 |
| | Health oriented education | 74.1 | 85 |
| | Other | 63.9 | 97 |
| | STEM | 80.9 | 94 |
| | Social Science | 76.8 | 69 |
| Serbian | Humanities | 90.4 | 313 |
| | STEM | 84.0 | 50 |
| | Social Science | 95.2 | 187 |
| Spanish | Humanities | 77.2 | 250 |
| | Health oriented education | 96.0 | 25 |
| | STEM | 88.0 | 25 |
| | Social Science | 89.6 | 250 |
| Tagalog | Humanities | 86.8 | 425 |
| | Other | 90.7 | 75 |
| Tamil | General knowledge | 70.6 | 500 |
| | STEM | 54.0 | 50 |
| Telugu | Applied Science | 73.5 | 166 |
| | Humanities | 66.0 | 191 |
| | Social Science | 66.9 | 166 |
| Turkish | Humanities | 62.0 | 166 |
| | Business & Commerce | 75.9 | 166 |
| | STEM | 52.0 | 50 |
| | Social Science | 62.0 | 166 |
| Ukrainian | Humanities | 92.4 | 250 |
| | STEM | 84.0 | 50 |
| | Social Science | 79.2 | 250 |
| Urdu | Humanities | 61.7 | 300 |
| | STEM | 63.3 | 49 |
| Uzbek | Humanities | 62.9 | 240 |
| | Other | 73.3 | 240 |
| | STEM | 84.0 | 50 |
| | Social Science | 71.4 | 21 |
| Vietnamese | Humanities | 88.0 | 250 |
| | STEM | 86.0 | 50 |
| | Social Science | 80.8 | 250 |

Table 9: GPT-4o (5-shot, In-language prompting) performance on INCLUDE-BASE per language and academic area. Areas with less than 30 examples were excluded from the analysis.

| Language | Academic Field | Regional Feature | Accuracy | Count |
|---|---|---|---|---|
| Albanian | History | Implicit | 93.1 | 58 |
| | Philosophy | Implicit | 97.6 | 82 |
| | Visual Arts | Implicit | 94.0 | 83 |
| | Business | Implicit | 85.7 | 223 |
| | Sociology | Implicit | 94.5 | 55 |
| Arabic | History | Implicit | 73.3 | 30 |
| | Language | Culture | 80.0 | 40 |
| | Accounting | Explicit | 89.5 | 57 |
| | Multiple exams | Implicit | 86.7 | 105 |
| | Driving License | Explicit | 76.2 | 105 |
| | Geography | Implicit | 65.3 | 49 |
| | Sociology | Implicit | 66.7 | 33 |
| Armenian | History | Culture | 26.3 | 95 |
| | History | Implicit | 41.1 | 95 |
| | Literature | Culture | 40.0 | 35 |
| | Driving License | Explicit | 72.2 | 79 |
| | Chemistry | Agnostic | 20.0 | 30 |
| | Geography | Implicit | 50.5 | 196 |
| Azerbaijani | Agriculture | Implicit | 85.3 | 34 |
| | Law | Explicit | 76.2 | 42 |
| | Management | Implicit | 66.7 | 36 |
| | Health | Implicit | 80.2 | 96 |
| | Economics | Implicit | 70.7 | 58 |
| Basque | Professional certification | Explicit | 64.8 | 500 |
| Belarusian | Language | Culture | 47.9 | 426 |
| | Literature | Culture | 67.4 | 43 |
| | Math | Agnostic | 40.8 | 49 |
| Bengali | Language | Culture | 62.5 | 40 |
| | Literature | Culture | 61.9 | 126 |
| | Multiple exams | Implicit | 80.1 | 166 |
| | Professional certification | Explicit | 84.3 | 166 |
| | Biology | Agnostic | 89.5 | 38 |
| Bulgarian | History | Implicit | 93.9 | 115 |
| | Philosophy | Implicit | 98.5 | 135 |
| | Geography | Implicit | 91.2 | 250 |
| Chinese | Medicine | Explicit | 57.1 | 35 |
| | Driving License | Explicit | 84.5 | 71 |
| | Professional certification | Explicit | 52.1 | 71 |
| | Political sciences | Implicit | 84.8 | 33 |
| Croatian | History | Implicit | 88.2 | 119 |
| | Philosophy | Implicit | 83.5 | 79 |
| | Religious Studies | Implicit | 90.2 | 51 |
| | Psychology | Implicit | 95.7 | 93 |
| | Sociology | Implicit | 94.8 | 135 |
| Dutch; Flemish | History | Culture | 89.4 | 141 |
| | Literature | Culture | 81.4 | 102 |
| | Economics | Implicit | 81.7 | 109 |
| | Geography | Implicit | 93.9 | 33 |
| | Sociology | Implicit | 90.1 | 101 |
| Estonian | Language | Culture | 89.1 | 147 |
| Finnish | Law | Explicit | 69.3 | 215 |
| | Economics | Implicit | 73.7 | 95 |
| | Political Sciences | Implicit | 61.5 | 96 |
| | Sociology | Implicit | 48.6 | 35 |
| French | Culturology | Culture | 94.8 | 77 |
| | Language | Culture | 79.0 | 124 |
| | Driving License | Explicit | 68.1 | 47 |
| | Geography | Implicit | 68.1 | 47 |
| Georgian | History | Implicit | 93.8 | 161 |
| | Language | Culture | 85.7 | 168 |
| | Law | Explicit | 83.6 | 171 |
| German | Geography | Implicit | 50.0 | 54 |
| Greek | Visual Arts | Implicit | 90.6 | 32 |
| | Management | Implicit | 89.1 | 64 |
| | Medical License | Explicit | 54.1 | 133 |
| | Professional Certification | Explicit | 60.9 | 133 |
| | Economics | Implicit | 85.8 | 120 |
| Hebrew | Logic | Agnostic | 60.0 | 50 |
| | Driving License | Explicit | 88.6 | 500 |
| Hindi | Education | Implicit | 84.3 | 70 |
| | History | Implicit | 86.7 | 30 |
| | Literature | Culture | 73.2 | 41 |
| | Multiple Exams | Implicit | 83.1 | 71 |
| | Medicine | Explicit | 91.5 | 71 |
| | Driving License | Explicit | 57.7 | 71 |
| | Professional Certification | Explicit | 70.4 | 71 |
| | Geography | Implicit | 75.0 | 48 |

Table 10: GPT-4o (5-shot, In-language prompting) performance on INCLUDE-BASE per language, academic field, and regional label. Fields with less than 30 examples were excluded from the analysis (Part 1)

| Language | Academic Field | Regional Feature | Accuracy | Count |
|---|---|---|---|---|
| Hungarian | Agriculture | Implicit | 82.4 | 170 |
| | Architecture and Design | Explicit | 85.7 | 42 |
| | Environmental Studies and Forestry | Implicit | 74.4 | 129 |
| | Economics | Implicit | 80.8 | 78 |
| | Geography | Implicit | 48.1 | 81 |
| Indonesian | Human Physical Performance and Recreation | Implicit | 71.2 | 125 |
| | Language | Culture | 79.5 | 78 |
| | Professional Certification | Region explicit | 83.2 | 125 |
| | Economics | Region explicit | 77.8 | 36 |
| | Geography | Implicit | 87.5 | 32 |
| | Sociology | Implicit | 87.7 | 57 |
| Italian | Agriculture | Implicit | 85.7 | 35 |
| | History | Implicit | 90.4 | 94 |
| | Professional Certification | Region explicit | 95.5 | 155 |
| | Psychology | Implicit | 95.0 | 60 |
| | Sociology | Implicit | 87.7 | 65 |
| Japanese | Driving License | Region explicit | 96.0 | 99 |
| | Medical License | Region explicit | 86.1 | 201 |
| | Professional Certification | Region explicit | 66.7 | 201 |
| Kazakh | History | Culture | 78.4 | 241 |
| | History | Implicit | 94.9 | 79 |
| | Literature | Culture | 76.7 | 180 |
| Korean | Professional Certification | Region explicit | 46.0 | 250 |
| | Economics | Implicit | 91.6 | 250 |
| Lithuanian | History | Implicit | 91.6 | 335 |
| | Finance | Implicit | 77.5 | 40 |
| | Professional Certification | Region explicit | 81.2 | 48 |
| | Earth Science | Agnostic | 97.1 | 34 |
| | Economics | Implicit | 93.5 | 77 |
| Malay | History | Implicit | 84.3 | 178 |
| | Accounting | Region explicit | 79.8 | 178 |
| | Geography | Implicit | 85.3 | 129 |
| Malayalam | History | Implicit | 61.5 | 52 |
| | Multiple Exams | Culture | 72.7 | 77 |
| | Health | Implicit | 55.0 | 100 |
| | Marine License | Explicit | 80.9 | 194 |
| Nepali | Driving License | Explicit | 83.2 | 250 |
| | Professional Certification | Explicit | 61.6 | 250 |
| North Macedonian | History | Implicit | 95.8 | 48 |
| | Philosophy | Implicit | 97.3 | 74 |
| | Visual Arts | Implicit | 97.1 | 102 |
| | Business | Implicit | 89.3 | 224 |
| | Sociology | Implicit | 92.5 | 53 |
| Persian | Literature | Culture | 51.6 | 31 |
| | Driving License | Explicit | 81.6 | 125 |
| | Professional Certification | Explicit | 43.2 | 125 |
| | Geography | Implicit | 66.0 | 47 |
| | Sociology | Implicit | 74.6 | 63 |
| Polish | Professional Certification | Explicit | 80.0 | 496 |
| | Math | Agnostic | 61.7 | 47 |
| Portuguese | Agriculture | Implicit | 70.0 | 40 |
| | Philosophy | Implicit | 83.3 | 84 |
| | Management | Implicit | 57.9 | 57 |
| | Health | Implicit | 70.3 | 37 |
| | Economics | Implicit | 89.7 | 126 |
| Russian | Education | Implicit | 87.0 | 69 |
| | Law | Explicit | 72.2 | 36 |
| | Management | Implicit | 66.2 | 65 |
| | Medicine | Explicit | 73.3 | 60 |
| | Marine License | Explicit | 56.5 | 69 |
| | Qualimetry | Explicit | 79.7 | 69 |
| | Economics | Implicit | 63.9 | 36 |
| Serbian | History | Implicit | 91.5 | 235 |
| | Philosophy | Implicit | 87.5 | 56 |
| | Psychology | Implicit | 99.2 | 125 |
| | Sociology | Implicit | 91.1 | 45 |
| Spanish | Language | Culture | 69.6 | 46 |
| | Law | Explicit | 67.0 | 109 |
| | Literature | Implicit | 93.8 | 64 |
| | Philosophy | Implicit | 90.3 | 31 |
| | Economics | Explicit | 95.6 | 91 |
| | Geography | Implicit | 86.2 | 159 |
| Tagalog | Culturology | Culture | 91.6 | 203 |
| | History | Culture | 85.3 | 116 |
| | Language | Culture | 79.2 | 106 |
| | Driving License | Explicit | 90.7 | 75 |
| Tamil | Multiple Exams | Implicit | 70.6 | 500 |
| Telugu | Education | Implicit | 73.0 | 100 |
| | History | Culture | 64.7 | 119 |
| | History | Implicit | 63.9 | 36 |
| | Economics | Explicit | 60.0 | 45 |
| | Geography | Implicit | 73.2 | 82 |
| | Political Sciences | Implicit | 63.3 | 30 |
| Turkish | History | Implicit | 71.2 | 73 |
| | Philosophy | Implicit | 74.6 | 63 |
| | Business | Implicit | 75.9 | 166 |
| | Geography | Implicit | 53.8 | 130 |
| | Sociology | Implicit | 91.7 | 36 |
| Ukrainian | Law | Explicit | 92.4 | 250 |
| | Physics | Agnostic | 84.0 | 50 |
| | Psychology | Implicit | 79.2 | 250 |
| Urdu | Culturology | Culture | 61.7 | 300 |
| Uzbek | History | Implicit | 66.1 | 124 |
| | Law | Explicit | 60.6 | 109 |
| | Medical License | Explicit | 73.3 | 240 |
| Vietnamese | History | Implicit | 88.3 | 239 |
| | Geography | Implicit | 80.8 | 250 |

Table 11: GPT-4o (5-shot, In-language prompting) performance on **INCLUDE-BASE** per language, academic field, and regional label. Fields with less than 30 examples were excluded from the analysis (Part 2)

| Language | Acc ($k$:50) | Acc ($k$:100) | Acc ($k$:200) | Acc ($k$:512) | Total gain |
|---|---|---|---|---|---|
| **Uzbek** | 51.4 | 60.6 | 66.6 | 68.6 | 17.2 |
| **Armenian** | 28.0 | 30.7 | 36.0 | 41.1 | 13.1 |
| **Malayalam** | 57.0 | 57.4 | 61.0 | 69.9 | 12.9 |
| **Urdu** | 53.7 | 56.8 | 58.8 | 62.2 | 8.5 |
| **Greek** | 58.0 | 58.2 | 63.8 | 66.4 | 8.4 |
| **Korean** | 60.4 | 61.0 | 62.4 | 68.8 | 8.4 |
| **Chinese** | 57.2 | 61.8 | 63.5 | 65.5 | 8.3 |
| **Finnish** | 63.3 | 64.4 | 67.0 | 69.1 | 5.8 |
| **Basque** | 60.0 | 60.8 | 63.8 | 64.8 | 4.8 |
| **Polish** | 74.1 | 75.2 | 75.4 | 78.1 | 4.0 |
| **Azerbaijani** | 67.7 | 69.2 | 70.4 | 71.5 | 3.8 |
| **Dutch; Flemish** | 81.9 | 82.9 | 83.8 | 85.3 | 3.4 |
| **Telugu** | 63.9 | 63.9 | 64.8 | 66.6 | 2.7 |
| **Hindi** | 72.0 | 72.4 | 73.7 | 74.4 | 2.4 |
| **German** | 64.0 | 65.5 | 65.5 | 66.2 | 2.2 |
| **Malay** | 80.6 | 81.8 | 82.4 | 82.8 | 2.2 |
| **Tamil** | 67.3 | 67.3 | 67.8 | 69.5 | 2.2 |
| **Arabic** | 76.3 | 76.8 | 77.9 | 78.4 | 2.1 |
| **russian** | 72.6 | 73.6 | 74.1 | 74.6 | 2.0 |
| **Italian** | 88.0 | 88.5 | 89.2 | 89.6 | 1.6 |
| **Spanish** | 82.4 | 83.1 | 83.3 | 84.0 | 1.6 |
| **Japanese** | 78.6 | 78.6 | 79.4 | 80.0 | 1.4 |
| **Georgian** | 86.2 | 86.4 | 87.0 | 87.6 | 1.4 |
| **Vietnamese** | 82.4 | 82.5 | 84.9 | 83.8 | 1.4 |
| **Turkish** | 63.5 | 64.1 | 64.4 | 64.8 | 1.3 |
| **Kazakh** | 79.2 | 79.6 | 80.4 | 80.4 | 1.2 |
| **Portuguese** | 72.8 | 73.5 | 73.5 | 74.0 | 1.2 |
| **Bengali** | 75.2 | 75.4 | 76.1 | 76.3 | 1.1 |
| **Persian** | 60.9 | 61.1 | 61.3 | 61.9 | 1.0 |
| **Belarusian** | 49.5 | 50.0 | 50.0 | 50.2 | 0.7 |
| **French** | 80.0 | 80.2 | 80.4 | 80.7 | 0.7 |
| **Indonesian** | 77.8 | 78.2 | 78.4 | 78.5 | 0.7 |
| **Albanian** | 88.9 | 89.3 | 89.3 | 89.5 | 0.6 |
| **Lithuanian** | 89.7 | 89.7 | 90.1 | 90.3 | 0.6 |
| **Estonian** | 92.0 | 92.0 | 92.4 | 92.4 | 0.4 |
| **Croatian** | 87.8 | 88.0 | 88.2 | 88.0 | 0.2 |
| **Hungarian** | 75.3 | 75.3 | 75.5 | 75.5 | 0.2 |
| **Nepali** | 71.8 | 72.0 | 71.6 | 72.0 | 0.2 |
| **Bulgarian** | 90.7 | 90.7 | 90.7 | 90.7 | 0.0 |
| **Hebrew** | 86.0 | 86.0 | 86.0 | 86.0 | 0.0 |
| **Macedonian** | 92.4 | 92.4 | 92.4 | 92.4 | 0.0 |
| **Serbian** | 91.5 | 91.5 | 91.5 | 91.5 | 0.0 |
| **Tagalog** | 87.4 | 87.4 | 87.4 | 87.4 | 0.0 |
| **Ukrainian** | 85.5 | 85.5 | 85.5 | 85.5 | 0.0 |

Table 12: **GPT-4o performance for different values of $k$ for in-language prompting** (the output generation length) per language on **INCLUDE-BASE** and total performance gain from $k$ = 50 to 512.

| Major training language | SoTA Monolingual | Monolingual Acc | GPT-4o | Qwen2.5-14B | Qwen2.5-7B |
|---|---|---|---|---|---|
| **Chinese** | **Baichuan-7B** | 38.7 | 68.1 | 82.2 | 78.3 |
| **Arabic** | **SILMA-9B-Instruct** | 56.9 | 78.1 | 70.5 | 61.6 |
| **Japanese** | **calm2-7b-chat** | 25.0 | 75.0 | 69.2 | 64.7 |
| **Korean** | **Korean-Mistral-Nemo-sft-dpo-12B** | 35.3 | 75.0 | 83.2 | 76.8 |
| **Russian** | **ruGPT-3.5-13B** | 53.8 | 69.0 | 68.2 | 59.6 |
| **German** | **SauerkrautLM-v2-14b-DPO** | 56.8 | 66.2 | 58.3 | 56.1 |

Table 13: Accuracy of the multilingual and monolingual models for answering **INCLUDE-BASE** questions for specific target languages.

| Language | $R^2$ | Model | $R^2$ |
|---|---|---|---|
| **Albanian** | 0.646 | **GPT-4o** | 0.077 |
| **Chinese** | 0.985 | **Qwen2.5-14B** | 0.546 |
| **French** | 0.770 | **Aya-expanse-32B** | 0.290 |
| **German** | 0.495 | **Aya-expanse-8B** | 0.333 |
| **Italian** | 0.953 | **Qwen2.5-7B** | 0.412 |
| **Lithuanian** | 0.945 | **Mistral-7B** | 0.231 |
| **Persian** | 0.833 | **Gemma-7B** | 0.001 |
| **Polish** | 0.831 | **Llama 3.1-70B** | 0.020 |
| **Portuguese** | 0.930 | **Llama 3.1-8B** | 0.001 |

Table 14: $R^2$ scores between the performance different models for newly-collected data and existing benchmarks stratified by language and model.

| Benchmark | Language coverage | Knowledge Coverage | Region agnostic (%) | Region related (%) |
|---|---|---|---|---|
| ArabicMMLU [1] | Arabic | Academic knowledge (elementary school, high school, university), Driving License | 24.8% | 75.2% |
| CMMLU [2] | Chinese | Academic knowledge (elementary school, high school, university) | 25.6% | 74.4% |
| PersianMMLU [3] | Persian | Academic knowledge (elementary school, high school, university) | 63.1% | 36.9% |
| TurkishMMLU [4] | Turkish | Academic knowledge (elementary school, high school, university) | 34.8% | 65.2% |
| VNHSGE [6] | Vietnamese | High school examinations | 40.4% | 59.6% |
| EXAMS [7] | 16 languages | High school examinations | 43.7% | 56.3% |
| **INCLUDE-BASE (ours)** | **44 languages** | **Academic knowledge (elementary school, high school, university), Professional examinations (Medical exam, Bar exam, Teaching exam), Occupational Licenses (Driving license, Marine license and more)** | 7.8% | **92.2%** |

Table 15: Existing published benchmarks descriptive and the comparison with **INCLUDE-BASE**.

| Error type | Description | Percentage of errors (%) |
|---|---|---|
| Computational Errors | Errors that occur when the model fails to perform arithmetic required to answer a question correctly. | 2.7% |
| Factual Knowledge Errors | Errors that involve the model lacking knowledge of facts unrelated to a specific region or language. For example, in the question, "Which of the following has a value between 1 and 1000? Choices: [Gini coefficient, base interest rate, personal credit score, corporate economic survey index]," the model may choose incorrectly due to a lack of factual knowledge. | 26.7% |
| Regional Knowledge Errors | Errors that arise when the model lacks knowledge specific to a particular region, even though it demonstrates proficiency in the language itself. | 38.6% |
| Format Errors | Errors that occur when the model fails to follow the prompt format or does not provide an answer, indicating challenges with language understanding or instruction processing. | 32.0% |

Table 16: Breakdown of error types across 150 manually-annotated examples from 6 languages.

