# OpenReview forum: "INCLUDE: Evaluating Multilingual Language Understanding with Regional Knowledge"
_ICLR.cc/2025/Conference — ICLR 2025 Spotlight_

### Official Review · Reviewer_i51G · 2024-10-28

**Soundness:** 4
**Presentation:** 4
**Contribution:** 4
**Rating:** 8
**Confidence:** 4

**Summary:**

The paper presents a new resource to evaluate the multilingualism of LLMs. The dataset, called Include, comprises multiple-choice questions from 44 languages about regional and global knowledge. The authors evaluate a battery of LLMs ranging from closed-source models (GPT-4o) to a variety of open-source models (C4AI-Aya, Mistral, Gemma, Qwen, Llama, XGLM, and Bloom). The paper presents a thorough analysis, comparing the performance of the models in different languages, domains, and regional/global knowledge. Finally, the authors discuss the challenges of evaluating these models in a setting with several languages.

**Strengths:**

* The paper is well-written and is easy to follow.
* The resource presented in this paper addresses a very prevalent challenge: the disparity in resources involved in the development of LLMs for English and non-English languages. Moreover, not only allows for the assessment of the quality of LLMs in the given languages but also in their regional cultural knowledge, which is most of the time ignored by the large body of research.

**Weaknesses:**

I did not find any major weaknesses in this paper. However, here are some possible improvements for this work:
* **Asses the difference in performance when the model is asked in English**: Etxaniz et al. (2024a) showed that these LLMs perform better in English although they understand the target language. They propose the " Self-translate " method, which prompts the same model to first translate the example into English before answering the question. Although I do not think you need to necessarily apply the "Self-translate" method, it would be nice to translate the examples into English (using machine translation) and compare the difference in performance. This way you can assess the knowledge encoded in the LLMs separately from their language capabilities.

**Questions:**

I have a couple of comments and questions about your work:
* **I missed some key references**: There are a couple of papers that are very aligned with this work, for instance, in the line of generating evaluation resources for LLMs in low-resource settings (particularly Basque):
  * (Etxaniz et al., 2024b): They present (along with pre-training data and models) an evaluation suite for Basque based on C1-level certificate of proficiency exams, reading comprehension exercises, trivia questions, and  Public Service examinations.
  * (Etxaniz et al., 2024c): They analyze the effect of regional vs non-regional knowledge of the LLM using an evaluation dataset that is parallel in English and Basque.
* **Have you tried any variants of Claude?** As for my experience, Claude performs better than GPT-4o in a couple of languages other than English that I tested. I am not asking to repeat the experiments with Claude, but I just have the curiosity of whether my personal feeling is reflected in your data.
* **Remove Answer Acc. from Table 1:** This metric is not comparable among the models in the table (as far as I understand). This can lead to confusion where worse models perform better just because they generate more wrongly formatted answers. If not removed, I would at least remove the bold results to avoid comparisons.


### References:
 * [Do Multilingual Language Models Think Better in English?](https://aclanthology.org/2024.naacl-short.46/) (Etxaniz et al., 2024a)
 * [Latxa: An Open Language Model and Evaluation Suite for Basque](https://aclanthology.org/2024.acl-long.799/) (Etxaniz et al., 2024b)
 * [BERTAQA: How Much Do Language Models Know About Local Culture?](https://arxiv.org/pdf/2406.07302) (Etxaniz et al., 2024c)

---

> ### Author Response · Authors · 2024-11-22
> **Response to Review**
>
> We thank the reviewer for their recognition of our work in addressing the prevalent challenge of resource disparity between English and non-English languages in LLM development and that the reviewer values our focus on assessing LLMs’ regional knowledge, an important but often overlooked aspect in the field. Finally, we appreciate that the reviewer did not find any major weaknesses in our work.
>
> **Additional experiments with English prompts:** The reviewer suggests exploring the effects of translating user prompts into English and comparing this approach with our existing experiments, where we prompt models using the native language of each sample. Based on the reviewer’s suggestions, we conducted new experiments where the user prompt instructions were translated into English while keeping the text of the question and the choices in the native language. We present new results for the most performant model variants within each model family, evaluated using both English and native language prompts:
>
> | Model                 | Native Language prompt | English Prompts |
> |-----------------------|------------------------|-----------------|
> | gpt-4o                | 77.3                   | 76.3            |
> | aya-expanse-32b       | 52.4                   | 56.0            |
> | llama3.1-70B-Instruct | 70.6                   | 70.7            |
> | aya-expanse-8b        | 37.1                   | 46.0            |
> | Mistral-7B            | 43.3                   | 44.9            |
> | Gemma-7B              | 54.5                   | 54.9            |
> | Qwen2.5-7B            | 54.1                   | 55.2            |
> | Qwen2.5-14B           | 61.4                   | 61.7            |
> | Llama-3.1-8B          | 51.0                   | 51.8            |
>
> _Table: Accuracy of the model for answering INCLUDE questions with user prompt instructions in the native language of each sample and with user prompt instructions in English._
>
> We expanded the analysis presented in the paper to include these new results, which suggest that English prompts can provide modest benefits for most models (with aya-expanse-8b and 32b being the exceptions), but that performance is generally within 1-2% of the performance of native language prompts.
>
> **Missing references:** As noted by the reviewer, we have included the referenced materials regarding generating evaluation resources for LLMs in low-resource settings in the updated version of the paper.
>
> **Changes in Table 1:** We will reformat the results in Table 1 to enhance clarity and improve reader comprehension and remove the bolded comparison for Answer Acc. At the same time, the reviewer brings up an interesting conceptual question! MCQ evaluations are proxy tasks for assessing the understanding that these models must display in open-ended, real-world interactions. While Total Acc. entangles the evaluation of regional knowledge understanding with other behaviors, such as outputting answers in the correct format, we think Answer Acc. provides an important comparison point where the model is only evaluated on examples where regional knowledge understanding errors can be disentangled from formatting errors.
>
> However, we also acknowledge that it does change the effective test set across models (as models will make formatting errors on different questions) and allows models to incidentally subvert the metric with incorrectly formatted answers, which may, in fact, indicate questions in which the model is less confident in a particular answer. We are happy to add this context to the paper as a footnote!
>
> **Experiments with Claude:** We would like to have results for the Claude model variants, but we are currently rate-limited by the API resources provided by Anthropic, preventing us from obtaining results across all settings in our paper. Once these results are available, we will add these models to our updated paper and launch a public leaderboard for INCLUDE.

---

> > ### Comment · Reviewer_i51G · 2024-11-22
> >
> > Thank you for your response. I appreciate the experiments conducted. I will keep my score, as I think it is high enough already.

---

### Official Review · Reviewer_E3PJ · 2024-10-30

**Soundness:** 3
**Presentation:** 3
**Contribution:** 3
**Rating:** 8
**Confidence:** 4

**Summary:**

This work identifies a lack of high-quality multilingual data available to evaluate LLMs. Existing resources are often translated from English and focus on a specific regions or domains. To address this gap, the authors introduce a large multi-lingual evaluation benchmark from close to 2000 regional exams across 44 languages, including high-, mid and low-resource languages. A range of 15 models is then evaluated on the new benchmark. The results indicate that performance largely varies across models and languages and there is still room for improvement even among the best performing models.

[see below for my response to the author comments.]

**Strengths:**

* The paper introduces a large and high-quality evaluation dataset that addresses previous short-comings of existing LLM benchmarks.
* Specifically, it identifies the fact that most existing multi-lingual resources are translated and therefore do not adequately reflect regional and cultural aspects that become relevant when models are actually deployed in respective environments.
* The paper’s structure is clear, the English is good and it is easy to follow
* The main contribution is an LLM benchmark, and as such it is not conceptually novel. However, it addresses a relevant short-coming and the introduction of „regional knowledge“ makes it original.
* Results are discussed in detail and Section 5 and the Appendix

**Weaknesses:**

* The paper argues that existing LLM benchmarks do not adequately cover multi-linguality nor regional knowledge and it lists related benchmarks in the introduction and related work. However, this argument is only on a qualitative level and would benefit from quantitative facts about, the size, number of languages, etc. of individual existing benchmarks. This could be included in a table providing an overview. [addressed]

* The paper is long on numbers, but short on more qualitative insights. For example, if a model fails to answer an exam question, is this due to a linguistic failure (the model fails to understand the question) or does it lack the regional/cultural knowledge? What types of errors do the models make? Insights of this kind would be crucial to make the results of the paper actionable for future model development, and while difficult to carry out automatically at scale, could be conducted manually on a sample for illustrative purposes. [substantially addressed]

* The evaluation also does not contain any hard evidence that the knowledge contained in the proposed benchmark cannot be tested by existing ones. It would be interesting to see correlations between performances on different benchmarks. This would be straightforward to do at least for the subsets of external benchmarks that are already included under „Rounding up the Benchmark“ in Section 3.1. [addressed]

**Questions:**

* On the technical side, it is not clear to me how exactly the model’s answer to a question is extracted from its output. In the first paragraph of Section 5.4 (lines 440-448), the authors mention that the adopt the prompt template by Hendrycks et al. and mention that the model does follow it consistently. What does the expected output look like and how exactly is the answer retrieved in case the model does not follow it? An example would be beneficial.

---

> ### Author Response · Authors · 2024-11-22
> **Response to Review [1/3]**
>
> We thank the reviewer for their thoughtful feedback and for recognizing the value of our dataset in reflecting authentic regional and cultural knowledge rather than relying on Western-centric datasets. We are also glad that the detailed discussion of results in Section 5 and the Appendix was found to be thorough and informative.
>
> **Existing Benchmarks Covering Regional Knowledge:** As discussed in the motivation and related work sections, there is currently no multilingual benchmark that offers both high language coverage and incorporates regional knowledge. Most existing benchmarks typically focus extensively on a single language across various (including regional) dimensions, or they cover a certain number of languages with large amounts of agnostic knowledge (i.e., translated versions of MMLU).
> As suggested by the reviewer, below, we provide more details on the list of existing benchmarks that were mentioned in the paper, which feature original content (not machine-translated), highlighting their language and knowledge coverage. As can be seen, INCLUDE contains a larger number of regional questions compared to other baselines. Furthermore, among regional categories, we note that most of these benchmarks contain large proportions of region-implicit questions, which is generally the weakest category of regionality (e.g., a question from a business school exam is considered region-implicit as regional perspectives may be reflected in the exam questions even if the subject material should be the same among different regions).
>
> | Benchmark | Language coverage | Knowledge Coverage                                                                                                                                                                                   | Region Agnostic | Region Explicit | Region Implicit | Cultural |
> |------|------|-------|------|----|---|----|
> | ArabicMMLU [1]  | Arabic | Academic knowledge (elementary, high school, university), Driving License                                                                                                                     | 24.3% | 11.0% | 53.2% | 11.5% |
> | CMMLU [2] | Chinese | Academic knowledge (elementary, high school, university)                                                                                                                                      | 25.6% | 23.0% | 38.3% | 13.1% |
> | PersianMMLU [3] | Persian | Academic knowledge (elementary, high school, university)                                                                                                                                      | 63.1% | 0.0% | 24.4% | 12.5% |
> | TurkishMMLU [4] | Turkish | Academic knowledge (elementary, high school, university)                                                                                                                                      | 45.0% | 0.0% | 45.2 | 9.8% |
> | VNHSGE [6] | Vietnamese | High school exams                                                                                                                                                                             | 40.4% | 0.0% | 59.6% | 0.0% |
> | EXAMS [7] | 16 languages | High school exams                                                                                                                                                                             | 43.7% | 8.3% | 47.5% | 0.5% |
> | **INCLUDE (ours)**  | **44 languages** | **Academic knowledge** (elementary, high school, university), **Professional examinations** (Medical exam, Bar exam, Teaching exam), **Occupational Licenses** (Driving license, Marine license and more) | 7.9%            | 29.2% | 48.1% | 14.8% |
>
> For now, we have added these results to Appendix A9 of the updated paper and will consider how to integrate this into the main body (without going over the page limit).
>
> >_References:_
> >
> >_[1]: Koto, Fajri, et al. "Arabicmmlu: Assessing massive multitask language understanding in arabic." arXiv preprint arXiv:2402.12840 >(2024)._
> >
> >_[2]: Li, Haonan, et al. "Cmmlu: Measuring massive multitask language understanding in chinese." arXiv preprint arXiv:2306.09212 (2023)._
> >
> >_[3]: Ghahroodi, O., et al. "Khayyam Challenge (PersianMMLU): Is Your LLM Truly Wise to The Persian Language? arXiv 2024." arXiv preprint arXiv:2404.06644._
> >
> >_[4]: Koto, Fajri, et al. "Large language models only pass primary school exams in Indonesia: A comprehensive test on IndoMMLU." arXiv preprint arXiv:2310.04928 (2023)._
> >
> >_[5]: Dao, Xuan-Quy, et al. "VNHSGE: VietNamese High School Graduation Examination Dataset for Large Language Models." arXiv preprint arXiv:2305.12199 (2023)._
> >
> >_[6]: Yüksel, Arda, et al. "Turkishmmlu: Measuring massive multitask language understanding in turkish." arXiv preprint arXiv:2407.12402 (2024)._
> >
> >_[7]: Hardalov, Momchil, et al. "EXAMS: A multi-subject high school examinations dataset for cross-lingual and multilingual question answering." arXiv preprint arXiv:2011.03080 (2020)._

---

> ### Author Response · Authors · 2024-11-22
> **Response to Review [2/3]**
>
> **Analysis of output errors on INCLUDE:**  As outlined in Section 5.3 of the paper, model performance—and consequently, the errors—are heavily influenced by the model's proficiency in the specific task and language. Following the reviewer's suggestion, we conducted a more detailed error analysis. We focused on six languages spanning high-resource (Chinese, Turkish), medium-resource (Bengali, Greek, Korean), and low-resource (Armenian) categories, selecting subject areas with the largest performance gaps compared to other subjects within the same language. For a breakdown of performance differences across subjects and languages, refer to Tables 6 and 7 in the Appendix. For the detailed analysis, we manually examined 150 examples, covering at least two subjects per language with 10 examples per subject, analyzing the questions and answers generated by GPT-4o.
> We observed four main types of errors: computational mistakes, factual mistakes, lack of regional knowledge, and model hallucinations. Each error type highlights distinct limitations in the model's capabilities, from arithmetic and factual knowledge to regional understanding and prompt adherence.
>
>
> | Error type                | Description | Percentage  of errors (%) |
> |---------------------------|----------------------------|---------------------------|
> | Computational Errors      | Errors that occur when the model fails to perform arithmetic required to answer a question correctly. | 2.7% |
> | Factual Knowledge  Errors | Errors that involve the model lacking knowledge of facts unrelated to a specific region or language. For example, in the question, `Which of the following has a value between 1 and 1000? Choices: [Gini coefficient, base interest rate, personal credit score, corporate economic survey index],` the model may choose incorrectly due to a lack of factual knowledge. | 26.7%                     |
> | Regional Knowledge Errors | Errors that arise when the model lacks knowledge specific to a particular region, even though it demonstrates proficiency in the language itself. | 38.6% |
> | Model Hallucinations      | Errors that occur when the model fails to follow the prompt format or does not provide an answer, indicating challenges with language understanding or instruction processing.                                                                                                                                                                                            | 32.0% |
>
> Our analysis revealed that the model's errors were distributed as follows: 38.6% were due to a lack of regional knowledge, 32% resulted from hallucinations, 26.7% were factual mistakes, and 2.7% were computational errors. We have added these results in Appendix A10 of the updated paper.
>
> In general, considering all models and their performance, we notice the following:
>
> - Output errors on the INCLUDE benchmark vary significantly based on model performance. High-performing models generally produce consistent output formats, whereas lower-performing models often generate inconsistent or hallucinated responses. In the most capable models, the primary error involves providing incorrect answers, often accompanied by flawed reasoning, as revealed in CoT experiments.
> - When analyzing errors by language resource levels, distinct patterns emerge: in high- and medium-resource languages, models often produce computational or knowledge-related errors and occasionally repeat the prompt in the 5-shot setting instead of answering the question. In low-resource languages, models frequently struggle with well-formatted responses and sometimes hallucinate information based on the demonstrations. These findings highlight the challenges of maintaining consistency and reasoning quality across languages with varying resource levels.

---

> ### Author Response · Authors · 2024-11-22
> **Response to Review [3/3]**
>
> **Limitations of the scope of existing benchmarks:** The INCLUDE benchmark makes three significant contributions: (1) introduces original datasets from languages that are either not covered or only partially covered by existing benchmarks, (2) leverages publicly available, knowledge-intensive multiple-choice question (MCQ) benchmarks in various languages, (3) organizes both the existing and newly introduced data under a unified taxonomy of knowledge, differentiating between regional knowledge and region-agnostic knowledge. In this context, INCLUDE incorporates existing datasets, which account for 39.8% of the total collected data and 31.7% of the INCLUDE-lite benchmark.
>
> To address the reviewer’s comment, we analyzed model performance on languages where both existing benchmark data and newly-collected data are available. We compared performance across these datasets and examined the correlation between them. Results were stratified by language and by model type. We visualized the performances using plots and calculated the $R^2$ scores to quantify the correlations. The updated paper includes these plots in the Appendix A9 in Figure 10, and the corresponding correlation scores are presented in the table below.
>
> | Language   | $R^2$ |   Model | $R^2$ |
> |--|--|--|--|
> | Albanian   | 0.646 |   GPT-4o | 0.077 |
> | Chinese    | 0.985 |  Qwen-2.5-14B    | 0.546 |
> | French     | 0.770 | Aya-expanse-32B | 0.290 |
> | German     | 0.495 | Aya-expanse-8B   | 0.333 |
> | Italian    | 0.953 | Qwen-2.5-7B  | 0.412 |
> | Lithuanian | 0.945 | Mistral-7B  | 0.231 |
> | Persian    | 0.833 | Gemma-7B  | 0.001 |
> | Polish     | 0.831 | Llama 3.1-70B  | 0.020 |
> | Portuguese | 0.930 | Llama 3.1-8B  | 0.001 |
>
> Overall, there is correlation between a model’s performance on a language-specific benchmark and its performance on the INCLUDE subset of that language, though with enough difference in most languages to justify the development of a new benchmark focused on regional knowledge. The analysis further reveals two key conclusions: (1) for a given language with multiple models having published performance data, a new model's performance range on a language-specific INCLUDE subset is generally predictable given a performance for that model on the same published benchmark in the same language (Language $R^2$ in table above). However, for a given model with published performance across different languages, its performance on INCLUDE cannot reliably be predicted when applied to a newly published language benchmark (Model $R^2$ in table above). This indicates that while INCLUDE may be less impactful for languages with existing published benchmarks, it is particularly valuable for assessing performance in languages with no prior resources, which is true for many of the languages in INCLUDE.
>
> **Answer extraction:** As highlighted in the paper and noted by the reviewer, our approach follows the prompting strategy outlined in the MMLU benchmark by Hendrycks et al. Specifically, we employ a 5-shot prompting method, with the prompts translated into the language corresponding to the question.
>
> We expect the model to follow the demonstration and output the number corresponding to the correct answer, similar to the behavior observed in English benchmarks. However, as discussed in Section 5.4, experimental results reveal that widely used prompting strategies in English benchmarks often fail to produce consistent outputs in multilingual settings. In some cases, the model's outputs vary depending on its capacity and proficiency in a particular language, potentially generating explanations, repeating the prompt, or providing definitions of answer choices—sometimes without outputting an answer at all. Below are examples that showcase different erroneous formats in generations (Outputs are translated from Armenian and Greek into English for presentation purposes):
>
> | Model | Output examples that do not follow the demonstrations (5-shots) |
> |---|-----|
> | GPT-4o | `Financial Management: 2` |
> | Aya-expanse-32B | `The correct answer is **2: The Rising of the Volunteers of 1792.**` |
> | Aya-expanse-7B  | `The "Einstein Tower" is an embodiment of:** - **Answer:** 3: expressionist architecture` |
>
> To ensure a fair evaluation of the models and accurately measure their ability to answer questions regardless of output format, we decided to extract the answer directly from the generated responses. This was achieved by manually inspecting a sample of questions for each language to identify patterns in the models' outputs. Based on these patterns, we designed a heuristic pipeline to systematically extract the answers.
>
> We also experimented with providing explicit instructions for the model to include the answer within a specific tag. However, this approach was unreliable, as some models in certain languages failed to follow the instructions, resulting in outputs without the specified tags.

---

> > ### Comment · Reviewer_E3PJ · 2024-11-25
> > **Thanks**
> >
> > Thank you for your detailed response. I appreciate these insights which go a long way towards addressing my concerns. Accordingly, I am increasing my score.

---

### Official Review · Reviewer_gceL · 2024-10-31

**Soundness:** 3
**Presentation:** 3
**Contribution:** 3
**Rating:** 5
**Confidence:** 4

**Summary:**

They construct a benchmark including 197,243 QA pairs in 44 languages. They collect it from local exam sources to measure the capabilities of multilingual LLMs in a variety of regional contexts. Experimental results show that these models perform inconsistently across different languages, particularly struggling with region-specific questions.

**Strengths:**

- The dataset is extensive, covering a broad range of domains and covering various application scenarios.

- The writing is generally clear.

**Weaknesses:**

- Experimental analysis is not quite enough and needs more takeaways.

Many of the experimental conclusions mentioned are known or intuitive. For example: In line 323 "Most instruction-tuned models perform slightly worse or on par with their base counterpart" which has been discovered in previous work[1][2] on English training corpus. Also for conclusions like LLMs' performance is poor on unseen language during pretraining.

Although a lot of effort has been put into building this benchmark, I feel that there is still a lack of takeaway, especially considering that one of the purposes of the Evaluation model is to improve the model. It's better to provide more find-gained analysis and insight like Figure 4 and also the insightful display (on the Main page rather than the Appendix) for conclusions like line142 model performs
worst on cultural questions.

- Doubts about the reliability of experimental findings

As you mentioned, some questions are "Cultural" and some are "Implicitly Regional". I am curious about the instructions for evaluating these two types of questions. Have you given the country or culture context of the questions being asked like: answer the question under  XXX culture?

- The Question Format is simply MCQ

Understanding that MCQ is more accessible and easier to deal with, however, open-ended questions can reflect the models' performance on real application scenarios.



[1] Urial: Tuning-Free Instruction Learning and Alignment for Untuned LLMs, 2023

[2] A+ B: A General Generator-Reader Framework for Optimizing LLMs to Unleash Synergy Potential, ACL, 2024

**Questions:**

Please refer to the Weaknesses.

---

> ### Author Response · Authors · 2024-11-21
> **Response to Review [1/2]**
>
> We thank the reviewer for their positive feedback and for highlighting the extensive coverage of domains and application scenarios in our dataset. We also appreciate the acknowledgment of the clarity of our writing.
>
> **Key takeaways around regional question types in INCLUDE:** The reviewer suggested conducting more fine-grained analyses of relative performance across region-explicit, region-agnostic, region-implicit, and cultural questions. We think this analysis would be beneficial too, and internally, have had long discussions about how fine-grained we could make such evaluations. However, due to the diverse exam sources, levels, and topics in our questions, direct comparisons of the relative "difficulty" of these regional question types might end up being misleading. To the reviewer’s point, while Figure 6 showed that models generally perform worse on cultural questions compared to region-agnostic ones, this could also be due to confounding factors (this is why Figure 6 was included in the appendix rather than highlighted in the main paper). For example, a collection of PhD-level cultural questions could simply be more challenging for models than elementary-level region-agnostic questions.
>
> In version 1 of INCLUDE, the distribution of difficulty levels across languages is imbalanced, complicating analytical comparisons across regional question types. Controlling for all of these factors this would lead to categories with limited examples when stratified by language, level, regionality, and topic. Consequently, we leave this to future work with an expanded data collection that enables rebalancing question types in INCLUDE to control for these factors. Instead, our current version prioritizes augmenting the evaluation of multilingual models with assessments of broad regional knowledge. Ultimately, the goal is for robust models to achieve high performance across all these regional dimensions.
>
> Despite this imbalance, we summarize the strong insights that still made it into our work:
> - Knowledge transfer is more effective between languages with shared scripts or typological similarities. However, when investigating models for which pre-training language mix is disclosed, we find they often perform unexpectedly well in untrained languages, potentially due to incidental data contamination during pre-training.
> - Lower performance in certain languages often relates to questions requiring regional knowledge, but this may also reflect the model's subject-specific capabilities (or the difficulty of the question) rather than regional knowledge alone, as regional labels are subject-dependent.
> - Models often struggle with non-English instructions, and instruction-tuning provides limited improvement, likely because most instruction-tuning data is predominantly in English.
> - When models are prompted with non-English data, they often fail to adhere to the exact format provided in the demonstrations. Our findings highlight the difficulty of standardizing evaluation for tasks that can produce varying output patterns, a challenge further amplified in multilingual evaluations.
>
> Based on these findings, we encourage LLM developers to evaluate model performance holistically by incorporating diverse geographic and cultural knowledge and increasing language diversity during instruction training, ensuring better local relevance for deployment contexts.

---

> ### Author Response · Authors · 2024-11-21
> **Response to Review [2/2]**
>
> **Regional and cultural dimensions of INCLUDE:** The reviewer suggests that prompting the model with the specific region the question is from as part of the context would improve performance for cultural and region-implicit questions. In our original experiments, we do not explicitly mention the country or cultural background of the exam topics for two reasons: (1) language often serves as a cultural proxy, allowing us to test models on regional knowledge implicitly (e.g., answering a question in Greek using Greek cultural knowledge). (2) users may not explicitly specify cultural context or origin in their queries, so we expect models to demonstrate regional knowledge based solely on the language of the prompt.
>
> While this approach motivates our work and experimental design, we incorporated the reviewer’s suggestion to conduct experiments that include the region and the language of each sample in the prompt instructions, asking the model to consider the cultural and linguistic nuances specific to that region. We present results for the best-performing large, medium, and small-scale models on specific language subsets across these two settings. We further stratify the results based on the type of knowledge: region agnostic or region related.
>
> | Model           | Original Prompt     |                 |                 |                 |          | Regional Prompt     |                 |                 |                 |          |
> |-----------------|---------------------|:---------------:|:---------------:|:---------------:|:--------:|---------------------|:---------------:|:---------------:|:---------------:|:--------:|
> |                 | Overall Performance | Region Agnostic | Region Explicit | Region Implicit | Cultural | Overall performance | Region Agnostic | Region Explicit | Region Implicit | Cultural |
> | gpt-4o          | 77.3                | 71.2            | 75.0            | 81.0            | 71.2     | 76.2                | 69.9            | 74.5            | 79.9            | 70.8     |
> | Aya-expanse-32B | 52.4                | 52.1            | 51.1            | 55.1            | 48.3     | 49.7                | 48.1            | 48.0            | 52.6            | 47.3     |
> | Qwen2.5-14B     | 61.4                | 61.2            | 60.5            | 64.9            | 53.6     | 61.1                | 60.9            | 59.7            | 64.7            | 52.2     |
> | Llama-3.1-8B    | 51.0                | 46.4            | 49.2            | 54.4            | 46.2     | 51.0                | 46.3            | 49.1            | 54.4            | 46.0     |
> | Qwen2.5-7B      | 54.1                | 54.7            | 53.6            | 56.7            | 46.1     | 54.0                | 54.4            | 53.4            | 56.5            | 46.7     |
>
> _Table: Accuracy of the models for answering INCLUDE questions for two prompting settings stratified by the regional feature._
>
> The results in the table show that providing explicit region and language information does not enhance model performance on regional questions. We have added these results to Appendix A8 of the updated paper. We encourage future INCLUDE users to experiment with different configurations and prompting strategies to further explore the extent of model performance.
>
> **MCQ format for automatic large-scale evaluation:** We agree that open-answer performance can provide different insights into model capabilities, including more nuanced assessment for real-world applications compared to MCQ. However, as the reviewer also points out, MCQ enables faster, closed-form evaluation while open-answer evaluation would require using human evaluators (considerably slowing the process of evaluation), or LLM-as-a-judge pipelines (which are likely less reliable in multilingual settings compared to English).  Our aim for this benchmark is to enable developers by providing an efficiently-evaluated, large-scale multilingual benchmark similar to widely used English benchmarks such as MMLU, MMMU, and MedQA (which are all MCQ as well).
>
> Finally, on an implementation level, we also note that each MCQ item in INCLUDE can also serve as an open-ended generation question by prompting the model without providing answer choices, allowing our benchmark to provide an opportunity for future work in multilingual LLM-as-a-judge open-answer evaluation methods.

---

> > ### Comment · Reviewer_gceL · 2024-11-22
> > **Following questions to Authors**
> >
> > Thank you for providing the response.
> >
> > My primary concern remains the reliability of the experimental findings. For instance, in Figure 1(a), it appears that asking questions without providing background context (indicating this is a Persian exam or Greek exam) introduces bias—such as the cultural preference differences observed between Persian and Greek cultures. While the authors argue that their approach of not providing specific instructions stems from the idea that “language often serves as a cultural proxy, allowing us to test models on regional knowledge implicitly,” I find this rationale unconvincing. In my view, such evaluation "Cultural" question and result will be ambiguous without providing the background context. The issue is not whether “prompting the model with the specific region the question is from as part of the context would improve performance for cultural knowledge”—a concern the authors seem to focus on—but rather the lack of clear instructions will affect the practical significance of the evaluation, which was why I inquired about the experimental instructions in the previous round.
> >
> > Although the authors have provided additional experiments, the results appear counterintuitive. Specifically, in the “Cultural” category, why does including a regional prompt result in lower performance compared to the “Original Prompt”? Is there any explanation for this discrepancy?

---

> > > ### Author Response · Authors · 2024-11-28
> > > **Response to follow up questions**
> > >
> > > We now better understand the concern regarding the lack of clear instructions potentially affecting the practical significance of the evaluation. While we understand this point, it is worth noting that this concern is not unique to our work; it is also common in English-language evaluations, where most widely used settings, such as those in Harness, HELM, and light-eval, typically provide very limited instructions. Consistent with this established practice, our study opted for minimal instructions to establish a simple baseline for performance. We also find it approximates a `user-focused’ setting more closely, where this information may not be explicitly mentioned in queries.
> > >
> > > However, we acknowledge the concern raised by the reviewer is valid and that practical assessment of regional knowledge understanding requires the region to be specified, particularly when the same language may be spoken across multiple regions, where cultural differences may persist despite a common language.
> > >
> > > Consequently, we have addressed the reviewers' recommendation to incorporate region-aware evaluations of all the models to enhance the practical significance of our results. To this end, we conducted additional experiments using region-aware prompts. The results and findings from these experiments have been included in the main results table (Table 1) of the updated version of the paper.
> > >
> > > As for why the regional prompts do not improve performance for Cultural questions, we have two potential hypotheses. First, many of the languages correspond to single regions (e.g., Estonian, Greek), so the language serves as an implicit indicator of the region. Second, in languages that may apply to multiple regions, most of our sources are from a single region (e.g., our French sources are from France, not Canada), which may be more dominantly represented in the model’s understanding or from the examples in the five-shot prompt. Consequently, the small drop may merely be empirical variance.

---

> > > > ### Author Response · Authors · 2024-12-03
> > > >
> > > > We would like to once again thank the reviewer for their valuable feedback during this rebuttal period that has allowed us to revise and improve our paper. Below is a summary of the key points discussed:
> > > >
> > > > - We clarified and summarized the main takeaways of our work, focusing on how models respond to the two key dimensions of INCLUDE: multilinguality and regional knowledge.
> > > >
> > > > - We explored the benefits and limitations of the multiple-choice question (MCQ) format, emphasizing INCLUDE's goal of providing developers with an efficiently evaluated, large-scale multilingual benchmark, comparable to widely used English benchmarks such as MMLU.
> > > >
> > > > - We revised the prompts to include regional information in both English and the native language of the samples and conducted a new set of experiments, which are now reported in Table 1.
> > > >
> > > > Please let us know if there is anything else we can address!

---

### Official Review · Reviewer_BvNN · 2024-11-04

**Soundness:** 3
**Presentation:** 3
**Contribution:** 2
**Rating:** 8
**Confidence:** 4

**Summary:**

This paper collects multiple-choice questions in a wide range of languages other than English and evaluates a selection of multilingual LLMs. For further analysis, the questions are categorized as cultural or region-specific (subdivided further into region-implicit and region-explicit). The evaluation shows that there is room for improvement in LLM performance and that the performance varies across different languages.

====
The score is increased after the rebuttal.

**Strengths:**

* The paper presents a new benchmark called Include, which collects multiple-choice questions sourced from existing exams in a variety of languages and domains. The selection of exams is conducted through a community survey, which appears to be an appropriate method for reaching out to native speakers and obtaining links from them. Two down-sampled versions of Include are available, which offer an equal amount of tasks in different languages and domains.

* The paper evaluates the performance of 15 multilingual LLMs in five-shot and zero chain-of-thought fashion.

* The paper analyses the results with respect to language, script, domain, and question category (cultural, region-implicit, region-explicit, academic) and formulates clear take-away messages.

**Weaknesses:**

* Only two LLMs are evaluated using both 5-shot and 0-shot CoT methods. It's unclear why the 0-shot CoT evaluation is missing for most models.

* Although the paper focuses on evaluating multilingual models, which seems to be a practical scenario, an ablation study with monolingual LLMs in a subset of languages could establish an upper bound for performance.

* The paper does not provide the actual names of the collected exams, along with their publication dates and formats. This information could be important for considering potential leakage to the LLM's pre-training data.

**Questions:**

* Why are only two LLMs evaluated using both 5-shot and 0-shot CoT methods?

* Have you try prompting the LLM in English and the target non-English question in 5-shot evaluation? Does the prompt language affect the performance?

* What is the chance that the LLMs have already been trained on some of the Include questions?

---

> ### Author Response · Authors · 2024-11-21
> **Response to Review [1/2]**
>
> We thank the reviewer for recognizing the value of INCLUDE as a benchmark encompassing regional knowledge across diverse languages and domains. We also appreciate the acknowledgment of our community-driven approach to selecting exams, which helps capture authentic linguistic and regional diversity, and are pleased that the reviewer found our analysis of multilingual LLM performance to have clear takeaways.
>
> **New Chain-of-Thought (CoT) experiments:** We only ran CoT in our initial paper for the larger GPT4o and Aya models as prior research indicates that small-scale language models (under 8 billion parameters) generally struggle to perform chain-of-thought (CoT) reasoning effectively [1,2,3]. However, based on the reviewer’s recommendation, we tested this setting on our benchmark and conducted CoT experiments on smaller (<= 8B parameters) and medium-sized models (14B parameters), selecting the most performant variants from the model families used in our initial experiments. The table below reports the accuracy of these models in both 5-shot and zero-shot CoT settings. Additionally, to expand the scope of our CoT experiments, we included the Llama 3.1-Instruct model with 70 billion parameters in our experimental process and analysis. For completion, we also include the results for aya-expanse-32B and GPT-4o models presented in our original submission.
>
> | Model           | Performance in 5-shot | Performance in zero-shot CoT |
> |-----------------|------------------------|------------------------------|
> | gpt-4o          | 77.3                   | 79.0                         |
> | aya-expanse-32b | 52.4                   | 51.4                         |
> | Llama3.1-70B    | 70.6                   | 60.6                         |
> | Qwen2.5-14B     | 61.4                   | 47.3                         |
> | Mistral-7B      | 43.3                   | 5.8                          |
> | Llama-3.1-8B    | 51.0                   | 9.1                          |
>
> _Table: Accuracy of the model for answering INCLUDE questions for 5-shot and zero-shot CoT settings._
>
> Our results align with findings in existing literature, confirming a fairly large performance gap between 5-shot and zero-shot CoT settings for smaller models. We have added these results to the paper in Table 1.
>
> >**_References:_**
> >
> >_[1]: Wei, Jason, et al. "Chain-of-thought prompting elicits reasoning in large language models." Advances in neural information processing systems 35 (2022): 24824-24837._
> >
> >_[2]: Mirzadeh, Iman, et al. "Gsm-symbolic: Understanding the limitations of mathematical reasoning in large language models." arXiv preprint arXiv:2410.05229 (2024)._
> >
> >_[3]: Wang, Yu, et al. “Strategic Chain-of-Thought: Guiding Accurate Reasoning in LLMs through Strategy Elicitation.” arXiv preprint arXiv:2409.03271 (2024). APA_
>
>
> **Experiments with English prompts:** The reviewer suggests prompting with English prompt instructions on non-English questions. In the submitted paper, we prompt models using the native language of each sample. Based on the reviewer’s suggestion, we conducted new experiments where the user prompt instructions were translated into English while keeping the text of the question and the choices in the native language. We present new results for the most performant model variants within each model family, evaluated using both English and native language prompts:
>
> | Model                 | Native Language prompt | English Prompts |
> |-----------------------|------------------------|-----------------|
> | GPT-4o                | 77.3                   | 76.3            |
> | Aya-expanse-32B       | 52.4                   | 56.0            |
> | Llama3.1-70B-Instruct | 70.6                   | 70.7            |
> | Aya-expanse-8B        | 37.1                   | 46.0            |
> | Mistral-7B            | 43.3                   | 44.9            |
> | Gemma-7B              | 54.5                   | 54.9            |
> | Qwen2.5-7B            | 54.1                   | 55.2            |
> | Qwen2.5-14B           | 61.4                   | 61.7            |
> | Llama-3.1-8B          | 51.0                   | 51.8            |
>
> _Table: Accuracy of the model for answering INCLUDE questions with user prompt instructions in the native language of each sample and with user prompt instructions in English._
>
> We expanded the analysis presented in the paper to include these new results, which suggest that English prompts can provide modest benefits for most models (with aya-expanse-8b and 32b achieving more significant increases), but that performance is generally within 1-2% of the performance of native language prompts. We added these results to Appendix A6.

---

> ### Author Response · Authors · 2024-11-21
> **Response to Review [2/2]**
>
> **Additional experiments on monolingual models:** The reviewer suggests that the performance of monolingual models for each language might provide an upper bound for expected performance of multilingual models. Based on data from open-LLM leaderboards for specific languages [1, 2, 3, 4], we observe that the top-performing models in each language are rarely monolingual models trained exclusively in that language, but rather multilingual models such as Qwen and Llama 3.1, so we do not believe that monolingual models would necessarily provide an upper bound on performance.
>
> Nevertheless, following the reviewer’s suggestion, we evaluated seven open-source monolingual models on the relevant language-specific subsets of INCLUDE, (i.e., the languages these models were pre-trained on). We compare their performance with the results of the most performant large-, medium-, and small-scale models on the specific language subsets. The results of this evaluation are presented in the table below.
>
> | Major Training Language | SoTA  Monolingual  Model | SoTA Monolingual Performance | Gpt-4o | Qwen2.5-14B | Qwen2.5-7B |
> |---|---|-|----|--|--|
> | Chinese  | Baichuan-7B | 38.7  | 68.1   | 82.2        | 78.3       |
> | Arabic   | SILMA-9B-Instruct | 56.9  | 78.1   | 70.5        | 61.6 |
> | Japanese | calm2-7b-chat | 25.0  | 75.0   | 69.2        | 64.7       |
> | Korean   | Korean-Mistral-Nemo-sft-dpo-12B | 35.3 | 75.0 | 83.2    | 76.8       |
> | Russian  | ruGPT-3.5-13B | 53.8  | 69.0   | 68.2 | 59.6 |
> | German   | SauerkrautLM-v2-14b-DPO  | 56.8  | 66.2   | 58.3 | 56.1 |
>
> _Table: Accuracy of the multilingual and monolingual models for answering INCLUDE questions for specific target languages._
>
> The table reveals that most monolingual models underperform the state-of-the-art small multilingual model Qwen-2.5 (7B), with the exception of the German monolingual model, SauerkrautLM-v2-14B-DPO, which performs on par with Qwen in the German language. We have added these results to Appendix A7 of the updated paper.
>
> >**_References:_**
> >
> >[1]: [Chinese Leaderboard](https://huggingface.co/spaces/BAAI/open_cn_llm_leaderboard)
> >
> >[2]: [Arabic Leaderboard](https://huggingface.co/spaces/OALL/Open-Arabic-LLM-Leaderboard)
> >
> >[3]: [Japanese Leaderboard](https://huggingface.co/spaces/llm-jp/open-japanese-llm-leaderboard)
> >
> >[4]: [Korean Leaderboard](https://huggingface.co/spaces/upstage/open-ko-llm-leaderboard)
>
> **Data contamination prevention:** The reviewer asks whether some of the tested LLMs may have already been trained on some of the questions in INCLUDE. As described in Section 3.1, INCLUDE is made up of a few previously-published benchmarks incorporated into INCLUDE, but also newly-collected exam materials from sources contributed by our multilingual community of native speakers. A significant portion of the newly-collected data was derived from PDFs and textbooks, which are less likely to have been included in models trained primarily on web-based data.
>
> Following the reviewer’s suggestion, we analyze the degree of contamination within the models using the mink%++ [1] method for training data detection in LLMs. This method determines whether an input next token forms a mode or has relatively high probability under the conditional categorical distribution. Using this scoring mechanism, one can predict if an input sequence is part of the model’s training data based on a decision threshold. This method achieves SOTA on the WikiMIA [2] benchmark for training data detection. We use the decision threshold that achieves the best performance on WikiMIA as the decision threshold for our analysis. Using this method, we computed the contamination rate for each language on four main-stream multilingual models: Aya-8B, XGLM-7B, LLaMA-3.1-8B, and Qwen-2.5-7B. We show the contamination rate results in the following table.
>
> |                                  | Aya-8B | XGLM-7B | Qwen-2.5-7B | LLaMA-3.1-8B |
> |----------------------------|------------|--------------|--------------------|---------------------|
> | Full Benchmark        | 0.02     | 0.17          | 0.13                | 0.29                 |
> | Newly collected   | 0.01     | 0.14          | 0.11                | 0.25                 |
>
> To further mitigate the risk of benchmark saturation as a result of data leakage when new models are trained, we have held back the complete dataset, comprising 197,243 entries. Instead, we will release these further questions and answers incrementally over the next year. We have also reserved a held-out dataset covering a wide range of the collected languages to be used for future experimental studies specifically aimed at analyzing data leakage over time.
>
> > _References:_
> >
> > _[1] Zhang, Jingyang, et al. "Min-k%++: Improved baseline for detecting pre-training data from large language models." arXiv preprint arXiv:2404.02936 (2024)._
> >
> > _[2] Shi, Weijia, et al. "Detecting pretraining data from large language models." arXiv preprint arXiv:2310.16789 (2023)._

---

> > ### Comment · Reviewer_BvNN · 2024-11-25
> > **Appreciate your response**
> >
> > I appreciate the author's response and think most of my concerns are resolved, specifically concerning model contamination. Thus, I am increasing my score by 1 point.

---

### Author Response · Authors · 2024-12-04
**Summary of discussion**

We sincerely thank all the reviewers for their valuable feedback and the time they dedicated to reviewing our work, INCLUDE. Below, we provide a summary of the main points discussed during the review period.

Overall, the feedback highlighted the dataset's comprehensiveness, authenticity, and relevance, along with the clarity and impact of the analysis. Specifically:
- The reviewers praised INCLUDE for its **extensive coverage of regional knowledge across languages and domains**, rather than relying on Western-centric paradigms.
- They also highlighted the **community-driven methodology** for selecting exams.
- The writing was noted for its **clarity and strong articulation**, while the analysis of multilingual LLM performance was appreciated for its clear and **actionable insights**.
- Furthermore, the thorough discussion of results in Section 5 and the Appendix was found to be both detailed and informative.

We also received constructive feedback from the reviewers, and we addressed it with new analysis and experiments, which we added to the paper and summarize below:

- **New Chain-of-Thought (CoT) experiments:** Based on reviewer BvNN’s recommendation to conduct CoT experiments on smaller models (<= 8B parameters), we added experiments on the most performant small variants from the model families used in our initial experiments. Our results align with findings in existing literature, confirming smaller models struggle in zero-shot CoT settings.

- **New experiments with English prompts:** Our initial prompts were in the same language as the evaluation questions. Following reviewer BVNN and i51G’s feedback, we conducted new experiments where the user prompt instructions were translated into English while keeping the question and choices in the native language. Our results suggest that English prompts can provide modest benefits for most models, but that performance is generally within 1-2% of the performance of native language prompts.

- **New experiments on monolingual models:** Based on reviewer BvNN’s question, we evaluated seven open-source monolingual models on the relevant language-specific subsets of INCLUDE, (i.e., the languages these models were pre-trained on). Most monolingual models underperform the state-of-the-art small multilingual model Qwen-2.5 (7B).

- **New experiments with region-aware prompts:** Based on reviewer gceL’s question, we conducted experiments that identify the region and language of each sample in the prompt instructions of the samples. The results show that providing explicit region and language information does not enhance model performance on regional questions.

- **Analysis on data contamination:** Based on reviewer BvNN’s question, we analyze the degree of contamination on four main-stream multilingual models using the mink%++ method for training data detection in LLMs, which showed small amounts of contamination among these models. To further mitigate the impact of benchmark leakage, we will release the complete dataset, comprising 197,243 entries, incrementally over the next year.

- **Analysis of output errors:**  Following reviewer E3PJ’s feedback, we conducted a more detailed error analysis of model outputs for INCLUDE, focusing on six languages (Chinese, Turkish, Bengali, Greek, Korean, Armenian). Our analysis revealed that the model's errors were distributed as follows: 38.6% were due to a lack of regional knowledge, 32% resulted from hallucinations, 26.7% were factual mistakes, and 2.7% were computational errors.

Finally, to further enhance the clarity and reproducibility of our work, we also provided the reviewers with additional details on the following points:
1. The methodology used for answer extraction in the evaluation process.
2. Clarifications regarding the existing benchmarks mentioned in the paper, specifically emphasizing those featuring original content (not machine-translated) and highlighting their coverage of languages and knowledge domains.
3. The limitations in the scope of existing benchmarks.

**All additional experiments, results, and analyses conducted for this rebuttal have been incorporated into the paper, either within the main sections or in the Appendix.**

---

### Meta-Review · Area_Chair_KkQn · 2024-12-20

**Metareview:**

This paper introduces a comprehensive multilingual evaluation benchmark called INCLUDE, comprising 197,243 multiple-choice questions across 44 languages. It was sourced from local exams to evaluate LLMs' capabilities in regional contexts. Experimental results show that current multilingual LLMs perform inconsistently across languages and struggle particularly with region-specific questions.

The benchmark is large-scale and incorporates authentic regional knowledge. Experimental analysis is conducted across multiple dimensions (language proficiency, regional knowledge types, prompting strategies).

The reviewers raise the issue of the lack of detailed comparison with existing benchmarks, the design of the problem format (only MCQ type questions), and also potential confounding factors in comparing difficulty across regional question types.

Overall I think the novelty of the newly introduced datasets is actually limited, with existing similar benchmarks already available. But I do agree with the reviewers that such large-scale resources might be useful for the community. Based on these considerations, I recommend accepting this paper.

**Additional Comments On Reviewer Discussion:**

The discussion period focused on several key points including benchmark comparisons, evaluation methodology, and regional knowledge assessment. The authors provided detailed responses and conducted additional experiments that addressed some of reviewer concerns, leading two reviewers to raise their scores.

---

### Decision · Program_Chairs · 2025-01-22

Accept (Spotlight)